# Diffusion Spectral Representation for Reinforcement Learning

**Dmitry Shribak**\*
Georgia Tech
shribak@gatech.edu

**Chen-Xiao Gao**\*
Nanjing University
gaocx@lamda.nju.edu.cn

**Yitong Li**
Georgia Tech
yli3277@gatech.edu

**Chenjun Xiao**
CUHK(SZ)
chenjunx@cuhk.edu.cn

**Bo Dai**
Georgia Tech
bodai@cc.gatech.edu

## Abstract

Diffusion-based models have achieved notable empirical successes in reinforcement learning (RL) due to their expressiveness in modeling complex distributions. Despite existing methods being promising, the key challenge of extending existing methods for broader real-world applications lies in the computational cost at inference time, *i.e.*, sampling from a diffusion model is considerably slow as it often requires tens to hundreds of iterations to generate even one sample. To circumvent this issue, we propose to leverage the flexibility of diffusion models for RL from a representation learning perspective. In particular, by exploiting the connection between diffusion models and energy-based models, we develop *Diffusion Spectral Representation (Diff-SR)*, a coherent algorithm framework that enables extracting sufficient representations for value functions in Markov decision processes (MDP) and partially observable Markov decision processes (POMDP). We further demonstrate how Diff-SR facilitates efficient policy optimization and practical algorithms while explicitly bypassing the difficulty and inference cost of sampling from the diffusion model. Finally, we provide comprehensive empirical studies to verify the benefits of Diff-SR in delivering robust and advantageous performance across various benchmarks with both fully and partially observable settings.

## 1 Introduction

Diffusion models have demonstrated remarkable generative modeling capabilities, achieving significant success in producing high-quality samples across various domains such as images and videos [Ramesh et al., 2021, Saharia et al., 2022, Brooks et al., 2024]. In comparison to other generative approaches, diffusion models stand out for their ability to represent complex, multimodal data distributions, a strength that can be attributed to two primary factors. First, diffusion models progressively denoise data by reversing a diffusion process. This iterative refinement process empowers them to capture complicated patterns and structures within the data distribution, thus enabling the generation of samples with unprecedented accuracy [Ho et al., 2020]. Second, diffusion models exhibit impressive mode coverage, effectively addressing a common issue of mode collapse encountered in other generative approaches [Song et al., 2020].

The potential of diffusion models is increasingly being investigated for sequential decision-making tasks. The inherent flexibility of diffusion models to accurately capture complex data distributions

---

\*Equal Contribution. Correspondence to: Bo Dai <bodai@cc.gatech.edu>

38th Conference on Neural Information Processing Systems (NeurIPS 2024).

makes them exceptionally suitable for both model-free and model-based methods in reinforcement learning (RL). There are three main approaches that attempts to apply diffusion models, including *diffusion policy* [Wang et al., 2022, Chi et al., 2023], *diffusion-based planning* [Janner et al., 2022, Jackson et al., 2024, Du et al., 2024], and *diffusion world model* [Ding et al., 2024, Rigter et al., 2023]. Empirical results indicate that diffusion-based approaches can stabilize the training process and enhance empirical performance compared with their conventional counterparts, especially in environments with high-dimensional inputs.

The flexibility of diffusion models, however, also comes with a substantial inference cost: generating even a single sample from a diffusion model is notably slow, typically requiring tens to thousands of iterations [Ho et al., 2020, Lu et al., 2022, Zhang and Chen, 2022, Song et al., 2023]. Furthermore, prior works also consider generating multiple samples to enhance quality, which further exacerbates this issue [Rigter et al., 2023]. The computational demands are particularly problematic for RL, since whether employing a diffusion-based policy or diffusion world model, the learning agent must frequently query the model for interactions with the environment during the learning phase or when deployed in the environment. This becomes the key challenge when extending diffusion-based methods for broader applications with more complex state spaces. Meanwhile, the planning with exploration issue has not been explicitly considered in the existing diffusion-based RL algorithms. The flexibility of diffusion models in fact induces extra difficulty in the implementation of the principle of optimism in the face of uncertainty in the planning step to balance the inherent trade-off between exploration vs. exploitation, which is indispensable in the online setting to avoid suboptimal policies [Lattimore and Szepesvári, 2020, Amin et al., 2021].

In conclusion, there has been insufficient work considering both efficiency and computation tractability for planning and exploration in a unified and coherent perspective, when applying diffusion model for sequential decision-making. This raises a very natural question, *i.e.*,

*Can we exploit the flexibility of diffusion models with efficient planning and exploration for RL?*

In this paper, we provide an **affirmative** answer to this question, based on our key observation that diffusion models, beyond their conventional role as generative tools, can play a crucial role in learning sufficient representations for RL. Specifically,

- By exploiting the energy-based model view of the diffusion model, we develop a coherent algorithmic framework *Diffusion Spectral Representation (Diff-SR)*, designed to learn representations that capture the latent structure of the transition function in Section 3.2;
- We then show that such diffusion-based representations are sufficiently expressive to represent the value function of any policy, which paves the way for efficient planning and exploration, circumventing the need for sample generation from the diffusion model, and thus, avoiding the inference costs associated with prior diffusion-based methods in Section 3.3;
- We conduct comprehensive empirical studies to validate the benefits of Diff-SR, in both fully and partially observable RL settings, demonstrating its robust, superior performance and efficiency across various benchmarks in Section 5.

## 2   Preliminaries

In this section, we briefly introduce the Markov Decision Processes, as the standard mathematical abstraction for RL, and diffusion models, as the building block for our algorithm design.

**Markov Decision Processes (MDPs).**   We consider *Markov Decision Processes* [Puterman, 2014] specified by the tuple $\mathcal{M} = \langle \mathcal{S}, \mathcal{A}, \mathbb{P}, r, \gamma, \mu_0 \rangle$, where $\mathcal{S}$ is the state space, $\mathcal{A}$ is the action space, $\mathbb{P} : \mathcal{S} \times \mathcal{A} \to \Delta(\mathcal{S})$ is the transition function, $r : \mathcal{S} \times \mathcal{A} \to \mathbb{R}$ is the reward function, $\gamma \in [0, 1)$ is the discount factor, $\mu_0 \in \Delta(\mathcal{S})$ is the initial state distribution[2]. The value function specifies the discounted cumulative rewards obtained by following a policy $\pi : \mathcal{S} \to \Delta(\mathcal{A})$, $V^\pi(s) = \mathbb{E}^\pi \left[ \sum_{t=0}^{\infty} \gamma^t r(s_t, a_t) | s_0 = s \right]$, where $\mathbb{E}^\pi$ denotes the expectation under the distribution induced by the interconnection of $\pi$ and the environment. The state-action value function is defined by

$$Q^\pi(s, a) = r(s, a) + \gamma \mathbb{E}_{s' \sim \mathbb{P}(\cdot|s,a)} \left[ V^\pi(s') \right].$$

The goal of RL is to find an optimal policy that maximizes the policy value, *i.e.*, $\pi^* = \operatorname{argmax}_\pi \mathbb{E}_{s \sim \mu_0}[V^\pi(s)]$.

---

[2]We use the standard notation $\Delta(\mathcal{X})$ to denote the set of probability distributions over a finite set $\mathcal{X}$

For any MDP, one can always factorize the transition operator through the singular value decomposition (SVD), *i.e.*,

$$\mathbb{P}(s'|s,a) = \langle \phi^*(s,a), \mu^*(s') \rangle \tag{1}$$

with $\langle \cdot, \cdot \rangle$ defined as the inner product. Yao et al. [2014], Jin et al. [2020], Agarwal et al. [2020a] considered a subset of MDPs, in which $\phi^*(s,a) \in \mathbb{R}^d$ with finite $d$, which is known as *Linear/Low-rank MDP*. The subclass is then generalized for infinite spectrum with fast decay eigenvalues [Ren et al., 2022a]. Leveraging this specific structure of the function class, this spectral view serves as an instrumental framework for examining the statistical and computational attributes of RL algorithms in the context of function approximation. In fact, the most significant advantage of exploiting said spectral structure is that we can represent its state-value function $Q^\pi(s,a)$ as a linear function with respect to $[r(s,a), \phi^*(s,a)]$ for *any policy* $\pi$,

$$Q^\pi(s,a) = r(s,a) + \gamma \mathbb{E}_{s' \sim \mathbb{P}(\cdot|s,a)} [V^\pi(s')] = r(s,a) + \left\langle \phi^*(s,a), \int_{\mathcal{S}} \mu^*(s') V^\pi(s') ds' \right\rangle. \tag{2}$$

It's important to highlight that in many practical scenarios, the feature mapping $\phi^*$ is often unknown. In the meantime, the learning of the $\phi^*$ is essentially equivalent to un-normalized conditional density estimation, which is notoriously difficult [LeCun et al., 2006, Song and Kingma, 2021, Dai et al., 2019]. Besides the optimization intractability in learning, the coupling of exploration to learning also compounds the difficulty: learning $\phi^*$ requires full-coverage data to capture $\mathbb{P}(s'|s,a)$, while the design of exploration strategy often relies on an accurate $\phi^*$ [Jin et al., 2020, Yang et al., 2020]. Recently, a range of spectral representation learning algorithms has emerged to address these challenges and provide an estimate of $\phi^*$ in both online and offline settings [Uehara et al., 2021]. However, existing methods either require designs of negative samples [Ren et al., 2022b, Qiu et al., 2022, Zhang et al., 2022], or rely on additional assumptions on $\phi^*$ [Ren et al., 2022c,a].

**Diffusion Models.** Diffusion models [Sohl-Dickstein et al., 2015, Ho et al., 2020, Song et al., 2020] are composed of a forward Markov process gradually perturbing the observations $x_0 \sim p_0(x)$ to a target distribution $x_T \sim q_T(x)$ with corruption kernel $q_{t+1|t}$, and a backward Markov process recovering the original observations distribution from the noisy $x_T$. After $T$ steps, the forward process forms a joint distribution,

$$q_{0:T}(x_{0:T}) = p_0(x_0) \prod_{t=0}^{T-1} q_{t+1|t}(x_{t+1}|x_t).$$

The reverse process can be derived by Bayes' rule from the joint distribution, *i.e.*,

$$q_{t|t+1}(x_t|x_{t+1}) = \frac{q_{t+1|t}(x_{t+1}|x_t) q_t(x_t)}{q_{t+1}(x_{t+1})},$$

with $q_t(x_t)$ as the marginal distribution at $t$-step. Although we can obtain the expression of reverse kernel $q_{t|t+1}(\cdot|\cdot)$ from Bayes's rule, it is usually intractable. Therefore, the reverse kernel is usually parameterized with a neural network, denoted as $p^\theta(x_t|x_{t+1})$. Recognizing that the diffusion models are a special class of latent variable models [Sohl-Dickstein et al., 2015, Ho et al., 2020], maximizing the ELBO emerges as a natural choice for learning,

$$\ell_{elbo}(\theta) = \mathbb{E}_{p_0(x)} \left[ D_{KL}(q_{T|0}||q_T) + \sum_{t=1}^{T-1} \mathbb{E}_{q_{t|0}} \left[ D_{KL}(q_{t|t+1,0}||p_{t|t+1}^\theta) \right] - \mathbb{E}_{q_{1|0}} \left[ \log p_{0|1}^\theta(x_0|x_1) \right] \right]$$

For continuous domain, the forward process of corruption usually employs Gaussian noise, *i.e.*, $q_{t+1|t}(x_{t+1}|x_t) = \mathcal{N}(x_{t+1}; \sqrt{1-\beta_{t+1}} x_t, \beta_{t+1} I)$, where $t \in \{0, \dots, T-1\}$. The kernel for backward process is also Gaussian and can be parametrized as $p^\theta(x_t|x_{t+1}) = \mathcal{N}\left( x_t; \frac{1}{\sqrt{1-\beta_{t+1}}} (x_{t+1} + \beta_{t+1} s^\theta(x_{t+1}, t+1)), \beta_{t+1} I \right)$, where $s^\theta(x_{t+1}, t+1)$ denotes the score network with parameters $\theta$. The ELBO can be specified as

$$\ell_{sm}(\theta) = \sum_{t=1}^{T} (1-\alpha_t) \mathbb{E}_{p_0} \mathbb{E}_{q_{t|0}} \left[ \left\| s^\theta(x_t, t) - \nabla_{x_t} \log q_{t|0}(x_t|x_0) \right\|^2 \right], \tag{3}$$

where $\alpha_t = \prod_{i=1}^{t} (1 - \beta_i)$. With the learned $s^\theta(x,t)$, the samples can be generated by sampling $x_T \sim \mathcal{N}(\mathbf{0}, \mathbf{I})$, and then following the estimated reverse Markov chain with $p^\theta(x_t|x_{t+1})$ iteratively. To ensure the quality of samples from diffusion models, the reverse Markov chain requires tens to thousands of iterations, which induces high computational costs for diffusion model applications.

**Random Fourier Features.** Random Fourier features [Rahimi and Recht, 2007, Dai et al., 2014] allow us to approximate infinite-dimensional kernels using finite-dimensional feature vectors. Bochner's theorem states that a continuous function of the form $k(x, y) = k(x - y)$ can be represented by a Fourier transform of a bounded positive measure [Bochner, 1932]. For Gaussian kernel $k(x - y)$, consider the feature $z_\omega(x) = \exp(-\mathbf{i}\omega^\top x)$, with $\omega \sim \mathcal{N}(0, I)$:

$$k(x - y) = \left[\int p(\omega)\exp(-\mathbf{i}\omega^\top(x - y))\mathrm{d}\omega\right] = \mathbb{E}_\omega[-\exp(\mathbf{i}\omega^\top(x - y))]]$$
$$= \mathbb{E}_\omega\left[z_\omega(x)z_\omega(y)^*\right] = \langle z_\omega(x), z_\omega(y)\rangle_{\mathcal{N}(\omega)}. \tag{4}$$

where $\langle \cdot, \cdot \rangle_{\mathcal{N}(\omega)}$ is a shorthand for $\mathbb{E}_{\omega\sim\mathcal{N}(0,I)}[\langle\cdot,\cdot\rangle]$. By sampling $\omega_1, \omega_2, \ldots, \omega_N \sim \mathcal{N}(0, I)$, we can approximate $k(x - y)$ with the inner product of finite-dimentional vectors $\hat{\boldsymbol{z}}_{\boldsymbol{\omega}}(x) = \frac{1}{\sqrt{N}}(z_{\omega_1}(x), z_{\omega_2}(x), \ldots, z_{\omega_N}(x))$ and $\hat{\boldsymbol{z}}_{\boldsymbol{\omega}}(y) = \frac{1}{\sqrt{N}}(z_{\omega_1}(y), z_{\omega_2}(y), \ldots, z_{\omega_N}(y))$.

# 3 Diffusion Spectral Representation for Efficient Reinforcement Learning

It is well known that the generation procedure of diffusion models becomes the major barrier for real-world application, especially in RL. Moreover, the complicated generation procedure makes the uncertainty estimation for exploration intractable. The spectral representation $\phi^*(s, a)$ provides an efficient way for planning and exploration, as illustrated in Section 2, which inspires our *Diffusion Spectral Representation (Diff-SR)* for RL, as our answer to the motivational question. As we will demonstrate below, the representation view of diffusion model enjoys the flexibility and also enables efficient planning and exploration, while directly bypasses the cost of sampling. For simplicity, we introduce Diff-SR in MDP setting in the main text. However, the proposed Diff-SR is also applicable for POMDPs as shown in Appendix B. We first illustrate the inherent challenges of applying diffusion models for representation learning in RL.

## 3.1 An Impossible Reduction to Latent Variable Representation [Ren et al., 2022a]

In the latent variable representation (LV-Rep) [Ren et al., 2022a], the latent variable model is exploited for spectral representation $\phi^*(s, a)$ in (1), by which an arbitrary state-value function can be linearly represented, and thus, efficient planning and exploration is possible. Specifically, in the LV-Rep, one considers the factorization of dynamics as

$$\mathbb{P}(s'|s, a) = \int p(z|s, a)p(s'|z)dz = \langle p(z|s, a), p(s'|z)\rangle_{L_2}. \tag{5}$$

By recognizing the connection between (5) and SVD of transition operator (1), the learned latent variable model $p(z|s, a)$ can be used as $\phi^*(s, a)$ for linearly representing the $Q^\pi$-function for an arbitrary policy $\pi$.

Since the diffusion model can be recast as a special type of latent variable model [Ho et al., 2020], the first straightforward idea is to extend LV-Rep with diffusion models for $p(z|s, a)$. We consider the following forward process that perturbs each $z_{i-1}$ with Gaussian noises:

$$p(z_i|z_{i-1}, s, a), \quad \forall i = 1, \ldots, k,$$

with $z_0 = s'$. Then, following the definition of the diffusion model, the backward process can be set as

$$q(z_{i-1}|z_i, s, a), \quad \forall i = 0, \ldots, k, \tag{6}$$

which are Gaussian distributions that denoise the perturbed latent variables. Thus, the dynamics model can be formulated as

$$\mathbb{P}(s'|s, a) = \int \prod_{i=2}^k q(z_{i-1}|z_i, s, a)q(s'|z_1, s, a)d\{z_i\}_{i=1}^k. \tag{7}$$

Indeed, Equation (7) converts the diffusion model to a latent variable model for dynamics modeling. However, the dependency of $(s, a)$ in $q(s'|z_1, s, a)$ correspondingly induces the undesirable dependency of $(s, a)$ into $\mu(s')$ in (1), and therefore the factorization provided by the diffusion model cannot linearly represent the $Q^\pi$-function — the vanilla LV-Rep reduction from diffusion models is impossible.

## 3.2 Diffusion Spectral Representation from Energy-based View

Instead of reducing to LV-Rep, in this work we extract the spectral representations by exploiting the relationship between diffusion models and energy-based models (EBMs).

**Spectral Representation from EBMs.** We parameterize the transition operator $\mathbb{P}(s'|s,a)$ using an EBM, *i.e.*,

$$\mathbb{P}(s'|s,a) = \exp\left(\psi(s,a)^\top \nu(s') - \log Z(s,a)\right), \quad Z(s,a) = \int \exp\left(\psi(s,a)^\top \nu(s')\right) ds'. \quad (8)$$

By simple algebra manipulation,

$$\psi(s,a)^\top \nu(s') = -\frac{1}{2}\left(\|\psi(s,a) - \nu(s')\|^2 - \|\psi(s,a)\|^2 - \|\nu(s')\|^2\right),$$

we obtain the quadratic potential function, leading to

$$\mathbb{P}(s'|s,a) \propto \exp\left(\|\psi(s,a)\|^2/2\right)\exp\left(-\|\psi(s,a) - \nu(s')\|^2/2\right)\exp\left(\|\nu(s')\|^2/2\right). \quad (9)$$

The term $\exp\left(-\frac{\|\psi(s,a) - \nu(s')\|^2}{2}\right)$ is the Gaussian kernel, for which we apply the random Fourier feature [Rahimi and Recht, 2007, Dai et al., 2014] and obtain the spectral decomposition of (8),

$$\mathbb{P}(s'|s,a) = \langle \phi_\omega(s,a), \mu_\omega(s')\rangle_{\mathcal{N}(\omega)}, \quad (10)$$

where $\omega \sim \mathcal{N}(0, I)$, and

$$\phi_\omega(s,a) = \exp\left(-\mathbf{i}\omega^\top \psi(s,a)\right)\exp\left(\|\psi(s,a)\|^2/2 - \log Z(s,a)\right), \quad (11)$$

$$\mu_\omega(s') = \exp\left(-\mathbf{i}\omega^\top \nu(s')\right)\exp\left(\|\nu(s')\|^2/2\right). \quad (12)$$

This bridges the factorized EBMs (8) to SVD, offering a spectral representation for efficient planning and exploration, as will be shown subsequently.

Exploiting the random feature to connect EBMs to spectral representation for RL was first proposed by Nachum and Yang [2021] and Ren et al. [2022c], but only Gaussian dynamics $p(s'|s,a)$ has been considered for its closed-form $Z(s,a)$ and tractable MLE. Equation (8) is also discussed in [Zhang et al., 2023b, Ouhamma et al., 2023, Zheng et al., 2022] for exploration with UCB-style bonuses. Due to the notorious difficulty in MLE of EBMs caused by the intractability of $Z(s,a)$ [Zhang et al., 2022], only special cases of (8) have been practically implemented. How to efficiently exploit the flexibility of general EBMs in practice still remains an open problem.

**Representation Learning via Diffusion.** We revisit the EBM understanding of diffusion models, which not only justifies the flexibility of diffusion models, but more importantly, paves the way for efficient learning of spectral representations (11) through Tweedie's identity for diffusion models.

Given $(s,a)$, we consider perturbing the samples from dynamics $s' \sim \mathbb{P}(s'|s,a)$ with Gaussian noise, *i.e.*, $\mathbb{P}(\tilde{s}'|s';\beta) = \mathcal{N}\left(\sqrt{1-\beta}s', \beta I\right)$. Then, we parametrize the corrupted dynamics as

$$\mathbb{P}(\tilde{s}'|s,a;\beta) = \int \mathbb{P}(\tilde{s}'|s';\beta)\mathbb{P}(s'|s,a)ds' \propto \exp\left(\psi(s,a)^\top \nu(\tilde{s}',\beta)\right), \quad (13)$$

where $\psi(s,a)$ is shared across all noise levels $\beta$, and $\mathbb{P}(\tilde{s}'|s,a;\alpha) \to \mathbb{P}(s'|s,a)$ with $\tilde{s}' \to s'$, as $\beta \to 0$, there is no noise corruption on $s'$.

**Proposition 1** (Tweedie's Identity [Efron, 2011]). *For arbitrary corruption $\mathbb{P}(\tilde{s}'|s';\beta)$ and $\beta$ in $\mathbb{P}(\tilde{s}'|s,a;\beta)$, we have*

$$\nabla_{\tilde{s}'} \log \mathbb{P}(\tilde{s}'|s,a;\beta) = \mathbb{E}_{\mathbb{P}(s'|\tilde{s}',s,a;\beta)}\left[\nabla_{\tilde{s}'} \log \mathbb{P}(\tilde{s}'|s';\beta)\right]. \quad (14)$$

This can be easily verified by simple calculation, *i.e.*,

$$\nabla_{\tilde{s}'} \log \mathbb{P}(\tilde{s}'|s,a;\beta) = \frac{\nabla_{\tilde{s}'}\mathbb{P}(\tilde{s}'|s,a;\beta)}{\mathbb{P}(\tilde{s}'|s,a;\beta)} = \frac{\nabla_{\tilde{s}'}\int \mathbb{P}(\tilde{s}'|s';\beta)\mathbb{P}(s'|s,a)ds'}{\mathbb{P}(\tilde{s}'|s,a;\beta)}$$

$$= \int \frac{\nabla_{\tilde{s}'} \log \mathbb{P}(\tilde{s}'|s';\beta)\mathbb{P}(\tilde{s}'|s';\beta)\mathbb{P}(s'|s,a)}{\mathbb{P}(\tilde{s}'|s,a;\beta)}ds' = \mathbb{E}_{\mathbb{P}(s'|\tilde{s}',s,a;\beta)}\left[\nabla_{\tilde{s}'} \log \mathbb{P}(\tilde{s}'|s';\beta)\right]. \quad (15)$$

For Gaussian perturbation with (13), Tweedie's identity (14) is applied as

$$\psi(s,a)^\top \nabla_{\tilde{s}'}\nu(\tilde{s}',\beta) = \mathbb{E}_{\mathbb{P}(s'|\tilde{s}',s,a;\beta)}\left[\frac{\sqrt{1-\beta}s' - \tilde{s}'}{\beta}\right]$$

$$\Rightarrow \tilde{s}' + \beta\psi(s,a)^\top \nabla_{\tilde{s}'}\nu(\tilde{s}',\beta) = \sqrt{1-\beta}\mathbb{E}_{\mathbb{P}(s'|\tilde{s}',s,a;\beta)}[s']. \quad (16)$$

Let $\zeta(\tilde{s}';\beta) = \nabla_{\tilde{s}'}\nu(\tilde{s}',\beta)$, we can learn $\psi(s,a)$ and $\zeta(\tilde{s}';\beta)$ by matching both sides of (16),

$$\min_{\psi,\zeta} \mathbb{E}_\beta\mathbb{E}_{(s,a,\tilde{s}')}\left[\left\|\tilde{s}' + \beta\psi(s,a)^\top \zeta(\tilde{s}',\beta) - \sqrt{1-\beta}\mathbb{E}_{\mathbb{P}(s'|\tilde{s}',s,a;\beta)}[s']\right\|^2\right] \quad (17)$$

---

**Algorithm 1** Diffusion Spectral Representation (Diff-SR) Training

---

1: **Input:** representation networks $\psi, \zeta$, noise levels $\{\beta^k\}_{k=1}^T$, replay buffer $\mathcal{D}$
2: **for** update step $= 1, 2, ..., N_{\text{rep}}$ **do**
3:      Sample a batch of $n$ transitions $\{(s_i, a_i, s_i')\}_{i=1}^n \sim \mathcal{D}$
4:      Sample noise schedules for each transition $\{\beta_i\}_{i=1}^n \sim \text{Uniform}(\beta^1, \beta^2, \ldots, \beta^T)$
5:      Corrupt the next states $\tilde{s}_i' \leftarrow \sqrt{1 - \beta_i} s_i' + \sqrt{\beta_i} \epsilon_i$, where $\epsilon_i \sim \mathcal{N}(0, I)$
6:      Optimize $\psi, \zeta$ via gradient descent by minimizing Eq (18)
7: **end for**
8: **Return** $\psi, \zeta$

---

which shares the same optimum of

$$\min_{\psi, \zeta} \ \ell_{\text{diff}}(\psi, \zeta) := \mathbb{E}_\beta \mathbb{E}_{(s,a,\tilde{s}',s')} \left[ \left\| \tilde{s}' + \beta \psi(s,a)^\top \zeta(\tilde{s}', \beta) - \sqrt{1 - \beta} s' \right\|^2 \right]. \tag{18}$$

The equivalence of (17) and (18) is provided in Appendix A.

**Diffusion Spectral Representation for $Q$-function.** The loss described by (18) estimates the score function $\psi(s,a)^\top \zeta(\tilde{s}', \beta)$ for diffusion models. In the context of generating samples, the score function suffices to drive the reverse Markov chain process. However, when deriving the random feature $\phi_\omega$ defined in (11), the partition function $Z(s,a)$ is indispensable. Furthermore, the random feature $\phi_\omega(s,a)$ is only conceptual with infinite dimensions where $\omega \sim \mathcal{N}(0, I)$. Next, we will proceed to analyze the structure of $Z(s,a)$ and finally construct the spectral representation with the learned $\psi(s,a)$.

We first illustrate $Z(s,a)$ is also linearly representable by random features of $\psi(s,a)$,

**Proposition 2.** *Denote $\rho_\omega(s,a) := \exp\left(-\mathbf{i}\omega^\top \psi(s,a) + \|\psi(s,a)\|^2/2\right)$, the partition function is linearly representable by $\rho_\omega(s,a)$, i.e., $Z(s,a) = \langle \rho_\omega(s,a), u \rangle_{\mathcal{N}(\omega)}$.*

*Proof.* We have $\phi_\omega(s,a) = \frac{\rho_\omega(s,a)}{Z(s,a)}$, and $\mathbb{P}(s'|s,a) = \left\langle \frac{\rho_\omega(s,a)}{Z(s,a)}, \mu_\omega(s') \right\rangle_{\mathcal{N}(\omega)}$, which implies

$$\int \mathbb{P}(s'|s,a)\, ds' = 1 \Rightarrow \left\langle \frac{\rho_\omega(s,a)}{Z(s,a)}, \underbrace{\int \mu_\omega(s')\, ds'}_{u} \right\rangle = 1 \Rightarrow \langle \rho_\omega(s,a), u \rangle_{\mathcal{N}(\omega)} = Z(s,a).$$

$\square$

Plug this into (11) and (2), we can represent the $Q^\pi$-function for arbitrary $\pi$ as

$$Q^\pi(s,a) = \left\langle \frac{\rho_\omega(s,a)}{\langle \rho_\omega(s,a), u \rangle}, \xi^\pi \right\rangle_{\mathcal{N}(\omega)} = \left\langle \underbrace{\exp\left(-\mathbf{i}\omega^\top \psi(s,a) - \log\left\langle \exp\left(-\mathbf{i}\omega^\top \psi(s,a)\right), u \right\rangle_{\mathcal{N}(\omega)}\right)}_{\varphi_{\omega,u}(s,a)}, \xi^\pi \right\rangle_{\mathcal{N}(\omega)}, \tag{19}$$

which eliminates the explicit partition function calculation.

We thereby construct *Diffusion Spectral Representation (Diff-SR)*, a tractable finite-dimensional representation, by approximating $\varphi_{\omega,u}(s,a)$ with some neural network upon $\psi(s,a)$. Specifically, in the definition of $\varphi_{\omega,u}(s,a)$ in (19), it contains Fourier basis $\exp\left(-\mathbf{i}\omega^\top \psi(s,a)\right)$, suggesting the use of trigonometry functions [Rahimi and Recht, 2007] upon $\psi(s,a)$, *i.e.*, $\sin\left(W_1^\top \psi(s,a)\right)$. Meanwhile, it also contains a product with $\frac{1}{\langle \exp(-\mathbf{i}\omega^\top \psi(s,a)), u \rangle_{\mathcal{N}(\omega)}}$, suggesting the additional non-linearity over $\sin\left(W_1^\top \psi(s,a)\right)$. Therefore, we consider the finite-dimensional neural network $\phi_\theta(s,a) = \texttt{elu}\left(W_2 \sin\left(W_1^\top \psi(s,a)\right)\right) \in \mathbb{R}^d$ with learnable parameters $\theta = (W_1, W_2)$, to approximate the infinite-dimensional $\varphi_\omega(s,a)$ with $\omega \sim \mathcal{N}(0, I)$.

**Remark (Connection to sufficient dimension reduction [Sasaki and Hyvärinen, 2018]):** The theoretical properties of the factorized potential function in EBMs (8) have been investigated in [Sasaki and Hyvärinen, 2018]. Specifically, the factorization is actually capable of universal approximation. Moreover, $\psi(s,a)$ also constructs an implementation of sufficient dimension reduction [Fukumizu et al., 2009], *i.e.*, given $\psi(s,a)$, we have $\mathbb{P}(s'|s,a) = \mathbb{P}(s'|\psi(s,a))$, or equivalently $s' \perp (s,a)|\psi(s,a)$. These properties justify the expressiveness and sufficiency of the learned $\psi(s,a)$. However, $\psi(s,a)$ in [Sasaki and Hyvärinen, 2018] is only estimated up to the partition function $Z(s,a)$, which makes it not directly applicable for planning and exploration in RL as we discussed.

---
**Algorithm 2** Online RL with Diff-SR
---
1: **Initialize** networks $\pi, (\xi_1, \theta_1), (\xi_2, \theta_2)$, and $\mathcal{D} = \emptyset$
2: **for** timestep $t = 1$ **to** $T$ **do**
3:      $a_t \sim \pi(\cdot|s_t)$
4:      $r_t = r(s_t, a_t),\ s'_t \sim \mathbb{P}(\cdot|s_t, a_t)$
5:      Compute bonus $b(s_t, a_t)$ using (23) (*Optional*)
6:      $\mathcal{D} \leftarrow \mathcal{D} \cup (s_t, a_t, r_t, s'_t)$
7:      Update $\psi$ with $\mathcal{D}$ by Algorithm 1
8:      Update the *critic* $(\xi_1, \theta_1), (\xi_2, \theta_2)$ by (21)
9:      Update the *policy* $\pi$ by $\max_\pi \mathbb{E}_{s \sim \mathcal{D}, a \sim \pi}[\min_{i \in \{1,2\}} Q_{\xi_i, \theta}(s, a)]$
10: **end for**
11: **Return** $\pi$.
---

## 3.3 Policy Optimization with Diffusion Spectral Representation

With the approximated finite-dimensional representation $\phi_\theta$, the $Q$-function can be represented as a linear function of $\phi_\theta$ and a weight vector $\xi \in \mathbb{R}^d$,

$$Q_{\xi, \theta}(s, a) = \phi_\theta(s, a)^\top \xi \tag{20}$$

This approximated value function can be integrated into any model-free algorithm for policy optimization. In particular, we update $(\xi, \theta)$ with the standard TD learning objective

$$\ell_{\text{critic}}(\xi, \theta) = \mathbb{E}_{s,a,r,s' \sim \mathcal{D}} \left[ \left( r + \gamma \mathbb{E}_{a' \sim \pi}[Q_{\bar{\xi}, \bar{\theta}}(s', a')] - Q_{\xi, \theta}(s, a) \right)^2 \right], \tag{21}$$

where $\pi$ is the learning algorithm's current policy, $(\bar{\xi}, \bar{\theta})$ is the target network, $\mathcal{D}$ is the replay buffer. We apply the *double Q-network trick* to stabilize training [Fujimoto et al., 2018]. In particular, two weights $(\xi_1, \theta_1), (\xi_2, \theta_2)$ are initialized and updated independently according to (21). Then the policy is updated by considering $\max_\pi \mathbb{E}_{s \sim \mathcal{D}, a \sim \pi}[\min_{i \in \{1,2\}} Q_{\xi_i, \theta_i}(s, a)]$. Algorithm 2 presents the pseudocode of online RL with Diff-SR. This learning framework is largely consistent with standard online reinforcement learning (RL) approaches, with the primary distinction being the incorporation of diffusion representations. As more data is collected, Line 7 specifies the update of the diffusion representation $\phi_\theta$ using Algorithm 1 and the latest buffer $\mathcal{D}$.

**Remark (Exploration with Diffusion Spectral Representation):** Following [Guo et al., 2023], even $\exp\left(-\mathbf{i}\omega^\top \psi(s, a)\right)$ is not enough for linearly representing $Q^\pi$, we still can exploit $\exp\left(-\mathbf{i}\omega^\top \psi(s, a)\right)$ for bonus calculation to implement the principle of optimism in the face of uncertainty for exploration in RL, *i.e.*,

$$b(s, a) = 1 - K^\top(s, a)(K + \lambda I)^{-1} K(s, a), \tag{22}$$

where $\mathcal{D}$ is a dataset of state action pairs, $k\left((s, a), (s', a')\right) := \exp\left(-\frac{\|\psi(s,a) - \psi(s',a')\|^2}{2}\right)$, $K(s, a) := [k((s, a), (s, a)_i)]_{(s,a)_i \in \mathcal{D}}$, and $K = [K((s, a)_j)]_{(s,a)_j \in \mathcal{D}}$. The bonus (22) derivation is straightforward by applying the connection between random feature and kernel, similar to [Zheng et al., 2022]. In practice, we can also approximate UCB bonus with Diff-SR as

$$b(s, a) = \phi_{\bar{\theta}}(s, a)^\top \left( \sum_{s', a' \in \mathcal{D}} \phi_{\bar{\theta}}(s, a) \phi_{\bar{\theta}}(s, a)^\top + \lambda I \right)^{-1} \phi_{\bar{\theta}}(s, a). \tag{23}$$

These bonuses are subsequently added to the reward in (21) to facilitate exploration.

**Remark (Comparison to Spectral Representation):** Both the proposed Diff-SR and the existing spectral representation for RL [Zhang et al., 2022, Ren et al., 2022b,a, Zhang et al., 2023b] are extracting the representation for $Q$-function. However, we emphasize that the major difference lies in that existing spectral representations seek low-rank linear representations, while our representation is sufficient for representing $Q$-function, but in a nonlinear form, as we revealed in (19). This nonlinearity in fact comes from the partition function $Z(s, a)$, which is constrained to be 1 in [Zhang et al., 2022, 2023b], and thus, less flexible for representation. Meanwhile, even with nonlinear representation, it has been shown that the corresponding bonus is still enough for exploration [Guo et al., 2023], without additional computational cost.

## 4 Related Work

**Representation learning in RL.** Learning good abstractions for the raw states and actions based on the structure of the environment dynamics is thought to facilitate policy optimization. To effectively capture the information in said dynamics, existing representation learning methods employ various techniques, such as reconstruction [Watter et al., 2015, Hafner et al., 2019b, Fujimoto et al., 2023], successor features [Dayan, 1993, Kulkarni et al., 2016, Barreto et al., 2017], bisimulation [Ferns et al., 2004, Gelada et al., 2019, Zhang et al., 2021], contrastive learning [Oord et al., 2018, Nachum and Yang, 2021], and spectral decomposition [Mahadevan and Maggioni, 2007, Wu et al., 2018, Duan et al., 2019, Ren et al., 2022b]. Previous works also leverage the assumption that the transition kernel possesses a low-rank spectral structure, which permits linear representations for the state-action value function and provably sample-efficient reinforcement learning [Jin et al., 2020, Yang and Wang, 2020, Agarwal et al., 2020b, Uehara et al., 2022]. Based on this, there have been several attempts towards both practical and theoretically grounded reinforcement learning algorithms by extracting the spectral representations from the transition kernel [Ren et al., 2022c, Zhang et al., 2022, Ren et al., 2022a, Zhang et al., 2023a]. Our approach aligns with this paradigm but distinguishes itself by learning these representations via diffusion and enjoying the flexibility of energy-based modeling.

**Diffusion model for RL.** By virtue of their ability to model complex and multimodal distributions, diffusion models present themselves as well-suited candidates for specific components in reinforcement learning. For example, diffusion models can be utilized to synthesize complex behaviors [Janner et al., 2022, Ajay et al., 2022, Chi et al., 2023, Du et al., 2024], represent multimodal policies [Wang et al., 2022, Hansen-Estruch et al., 2023, Chen et al., 2023b], or provide behavior regularizations [Chen et al., 2023a]. Another line of research utilizes the diffusion model as the world model. Among them, DWM [Ding et al., 2024] and SynthER [Lu et al., 2023] train a diffusion model with off-policy dataset and augment the training dataset with synthesized data, while Poly-GRAD [Rigter et al., 2023] and PGD [Jackson et al., 2024] sample from the diffusion model with policy guidance to generate near on-policy trajectory. On a larger scale, UniSim [Yang et al., 2023] employs a video diffusion model to learn a real-world simulator that accommodates instructions in various modalities. All of these methods incur great computational costs because they all involve iteratively sampling from the diffusion model to generate actions or trajectories. In contrast, our method leverages the capabilities of diffusion models from the perspective of representation learning, thus circumventing the generation costs.

## 5 Experiments

We evaluate our method with state-based MDP tasks (Gym-MuJoCo locomotion [Todorov et al., 2012]) and image-based POMDP tasks (Meta-World Benchmark [Yu et al., 2020]) in this section. Besides, we also provide experiments with state-based POMDP tasks in Appendix E. Our code is publicly released at the project website.

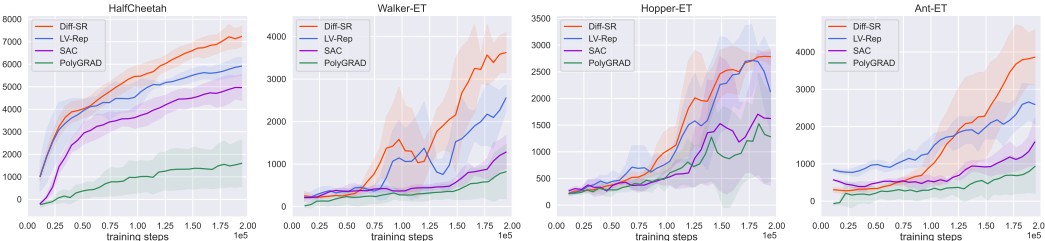

Figure 1: The performance curves of the Diff-SR and baseline methods on MBBL tasks. We report the mean (solid line) and one standard deviation (shaded area) across 4 random seeds.

### 5.1 Results of Gym-MuJoCo Tasks

In this first set of experiments, we compare Diff-SR against both model-based and model-free baseline algorithms, using the implementations provided by MBBL [Wang et al., 2019]. For representation-based RL algorithms, we include LV-Rep [Ren et al., 2022a], SPEDE [Ren et al., 2022c] and Deep

Table 1: Performances of Diff-SR and baseline RL algorithms after 200K environment steps. Results are averaged across 4 random seeds and a window size of 10K steps. Results marked with * are taken from MBBL [Wang et al., 2019] and † are taken from LV-Rep [Ren et al., 2022a].

|  |  | HalfCheetah | Reacher | Humanoid-ET | Pendulum | I-Pendulum |
|---|---|---|---|---|---|---|
| Model-Based RL | ME-TRPO* | 2283.7 ± 900.4 | −13.4 ± 5.2 | 72.9 ± 8.9 | **177.3 ± 1.9** | −126.2 ± 86.6 |
|  | PETS-RS* | 966.9 ± 471.6 | −40.1 ± 6.9 | 109.6 ± 102.6 | 167.9 ± 35.8 | −12.1 ± 25.1 |
|  | PETS-CEM* | 2795.3 ± 879.9 | −12.3 ± 5.2 | 110.8 ± 90.1 | 167.4 ± 53.0 | −20.5 ± 28.9 |
|  | Best MBBL* | 3639.0 ± 1135.8 | **−4.1 ± 0.1** | 1377.0 ± 150.4 | **177.3 ± 1.9** | **0.0 ± 0.0** |
|  | PolyGRAD | 2563.5 ± 204.2 | −20.7 ± 1.9 | 1026.6 ± 58.7 | 166.3 ± 6.3 | −3.5 ± 4.8 |
| Model-Free RL | PPO* | 17.2 ± 84.4 | −17.2 ± 0.9 | 451.4 ± 39.1 | 163.4 ± 8.0 | −40.8 ± 21.0 |
|  | TRPO* | −12.0 ± 85.5 | −10.1 ± 0.6 | 289.8 ± 5.2 | 166.7 ± 7.3 | −27.6 ± 15.8 |
|  | SAC* (3-layer) | 4000.7 ± 202.1 | −6.4 ± 0.5 | **1794.4 ± 458.3** | 168.2 ± 9.5 | −0.2 ± 0.1 |
| Representation RL | DeepSF† | 4180.4 ± 113.8 | −16.8 ± 3.6 | 168.6 ± 5.1 | 168.6 ± 5.1 | −0.2 ± 0.3 |
|  | SPEDE† | 4210.3 ± 92.6 | −7.2 ± 1.1 | 886.9 ± 95.2 | 169.5 ± 0.6 | 0.0 ± 0.0 |
|  | LV-Rep† | 5557.6 ± 439.5 | −5.8 ± 0.3 | 1086 ± 278.2 | 167.1 ± 3.1 | **0.0 ± 0.0** |
|  | **Diff-SR** | **7223.53 ± 437.1** | −6.5 ± 2.0 | 821.3 ± 118.9 | 162.3 ± 3.0 | 0.0 ± 0.1 |
|  |  | Ant-ET | Hopper-ET | S-Humanoid-ET | CartPole | Walker-ET |
| Model-Based RL | ME-TRPO* | 42.6 ± 21.1 | 1272.5 ± 500.9 | −154.9 ± 534.3 | 160.1 ± 69.1 | −1609.3 ± 657.5 |
|  | PETS-RS* | 130.0 ± 148.1 | 205.8 ± 36.5 | 320.7 ± 182.2 | 195.0 ± 28.0 | 312.5 ± 493.4 |
|  | PETS-CEM* | 81.6 ± 145.8 | 129.3 ± 36.0 | 355.1 ± 157.1 | 195.5 ± 3.0 | 260.2 ± 536.9 |
|  | Best MBBL* | 275.4 ± 309.1 | 1272.5 ± 500.9 | **1084.3 ± 77.0** | **200.0 ± 0.0** | 312.5 ± 493.4 |
|  | PolyGRAD | 433.6 ± 158.6 | 1151.6 ± 182.3 | **1110.1 ± 181.8** | 193.4 ± 11.5 | 268.4 ± 77.3 |
| Model-Free RL | PPO* | 80.1 ± 17.3 | 758.0 ± 62.0 | 454.3 ± 36.7 | 86.5 ± 7.8 | 306.1 ± 17.2 |
|  | TRPO* | 116.8 ± 47.3 | 237.4 ± 33.5 | 281.3 ± 10.9 | 47.3 ± 15.7 | 229.5 ± 27.1 |
|  | SAC* (3-layer) | 2012.7 ± 571.3 | 1815.5 ± 655.1 | 834.6 ± 313.1 | **199.4 ± 0.4** | 2216.4 ± 678.7 |
| Representation RL | DeepSF† | 768.1 ± 44.1 | 548.9 ± 253.3 | 533.8 ± 154.9 | 194.5 ± 5.8 | 165.6 ± 127.9 |
|  | SPEDE† | 806.2 ± 60.2 | 732.2 ± 263.9 | 986.4 ± 154.7 | 138.2 ± 39.5 | 501.6 ± 204.0 |
|  | LV-Rep† | 2511.8 ± 460.0 | 2204.8 ± 496.0 | 963.1 ± 45.1 | **200.7 ± 0.2** | 2523.5 ± 333.9 |
|  | **Diff-SR** | **4788.6 ± 623.1** | **2800.5 ± 95.4** | 1160.3 ± 150.1 | 199.9 ± 1.1 | **3722.2 ± 406.1** |

Successor Feature (DeepSF) [Barreto et al., 2017] as baselines. Note that the SPEDE is a special case of Gaussian EBM. We also include PolyGRAD [Rigter et al., 2023], a recent method that utilizes diffusion models for RL as an additional baseline. All algorithms are executed for 200K environment steps and we report the mean and standard deviation of performances across 4 random seeds. More implementation details including the hyper-parameters can be found in Appendix C.

Table 1 presents the results, demonstrating that Diff-SR achieves significantly better or comparable performance to all baseline methods in most tasks except for Humanoid-ET. Specifically, in Ant and Walker, Diff-SR outperforms the second highest baseline LV-Rep by 90% and 48%. Moreover, Diff-SR consistently surpasses PolyGRAD in nearly all environments. We also provide the learning curves of Diff-SR and baseline methods in Figure 1 for an illustrative interpretation of the sample efficiency Diff-SR brings.

**Computational Efficiency and Runtime Comparison**
Compared to other diffusion-based RL algorithms, Diff-SR harnesses diffusion's flexibility while circumventing the time-consuming sampling process. To showcase this, we record the runtime of Diff-SR and PolyGRAD on MBBL tasks using workstations equipped with Quadro RTX 6000 cards. Results in Figure 2 illustrate that Diff-SR is about 4× faster than PolyGRAD, and such advantage is consistent across all environments. We provide a per-task breakdown of the runtime results in Appendix 4 due to space constraints.

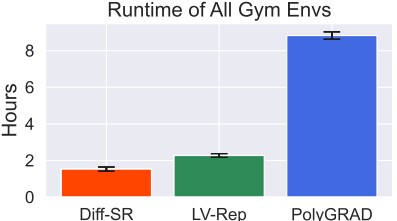

Figure 2: Runtime comparison between Diff-SR vs. LV-Rep vs. diffusion-based RL (PolyGRAD).

## 5.2 Results of Meta-World Tasks

As the most difficult setting, we evaluate Diff-SR with 8 visual-input tasks selected from the Meta-World Benchmark. Rather than directly diffuse over the space of raw pixels, we resort to techniques similar to the Latent Diffusion Model (LDM) [Rombach et al., 2022] which first encodes the raw

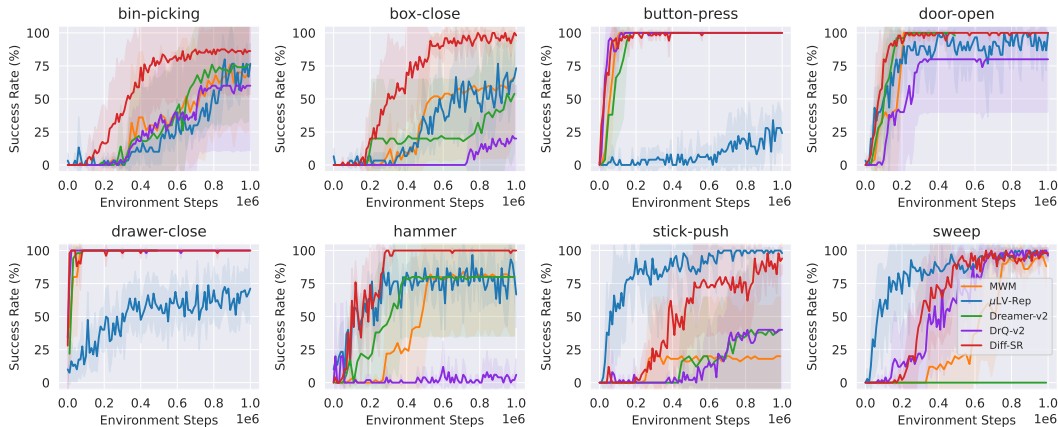

Figure 3: Performance curves for image-based POMDP tasks from Meta-World. We report the mean (solid line) and the standard deviation (shaded area) of performances across 5 random seeds.

observation image to a 1-D compact latent vector and afterward performs the diffusion process over the latent space. To deal with the partial observability, we truncate the history frames with length $L$, encode these $L$ frames into their latent embeddings, concatenate them together, and treat them as the state of the agent. The learning of diffusion representation thus translates into predicting the next frame's latent embedding with the action and $L$-step concatenated embedding. Following existing practices [Zhang et al., 2023a], we set $L = 3$ for all Meta-World tasks. We use DrQ-V2 [Yarats et al., 2021] as the base RL algorithm, and more details of the implementations are deferred to Appendix F.

The performance curves of Diff-SR and the baseline algorithms are presented in Figure 3. We see that Diff-SR achieves a greater than 90% success rate for seven of the tasks, 4 more tasks than the second best baseline $\mu$LV-Rep. Overall, Diff-SR exhibits superior performance, faster convergence speed, and stable optimization in most of the tasks compared to the baseline methods. Finally, although Diff-SR does not require sample generation, we present the reconstruction results to validate the efficacy of the score function $\psi\left(s, a\right)^{\top} \zeta\left(\tilde{s}', \beta\right)$ in Figure 6.

# 6  Conclusions and Discussions

We introduce Diff-SR, a novel algorithmic framework designed to leverage diffusion models for reinforcement learning from a representation learning perspective. The primary contribution of our work lies in exploiting the connection between diffusion models and energy-based models, thereby enabling the extraction of spectral representations of the transition function. We demonstrate that such diffusion-based representations can sufficiently express the value function of any policy, facilitating efficient planning and exploration while mitigating the high inference costs typically associated with diffusion-based methods. Empirically, we conduct comprehensive studies to validate the effectiveness of Diff-SR in both fully and partially observable sequential decision-making problems. Our results underscore the robustness and advantages of Diff-SR across various benchmarks. However, the main limitation of this study is that Diff-SR has not yet been evaluated with real-world and multi-task data. Future work will focus on testing Diff-SR on real-world applications, such as robotic control.

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

# A Detailed Derivations

**Proposition 3.** *Equation* (17) *shares the same optimal solutions with Equation* (18).

*Proof.* Denote $b = \sqrt{1-\beta}\mathbb{E}_{\mathbb{P}(s'|\tilde{s}',s,a;\beta)}[s']$ for simplicity, and we have

$$\mathbb{E}_\beta \mathbb{E}_{(s,a,\tilde{s}',s')}\left[\left\|\tilde{s}' + \beta\psi(s,a)^\top \zeta(\tilde{s}',\beta) - \sqrt{1-\beta}s'\right\|^2\right]$$

$$= \mathbb{E}_\beta \mathbb{E}_{(s,a,\tilde{s}',s')}\left[\left\|\tilde{s}' + \beta\psi(s,a)^\top \zeta(\tilde{s}',\beta) - b + b - \sqrt{1-\beta}s'\right\|^2\right]$$

$$= \mathbb{E}_\beta \mathbb{E}_{(s,a,\tilde{s}',s')}\left[\left\|\tilde{s}' + \beta\psi(s,a)^\top \zeta(\tilde{s}',\beta) - b\right\|^2\right]$$

$$+ \underbrace{2\,\mathbb{E}_\beta \mathbb{E}_{(s,a,\tilde{s}',s')}\left[\left(\tilde{s}' + \beta\psi(s,a)^\top \zeta(\tilde{s}',\beta) - b\right)^\top \left(b - \sqrt{1-\beta}s'\right)\right]}_{0}$$

$$+ \underbrace{(1-\beta)\,\mathbb{E}_\beta \mathbb{E}_{(s,a,\tilde{s}',s')}\left[\left\|s' - \mathbb{E}_{\mathbb{P}(s'|\tilde{s}',s,a;\beta)}[s']\right\|^2\right]}_{\texttt{constant}}.$$

The second term equals to 0 comes from the definition of $b$. Moreover, since $b$ is independent w.r.t. $s'$, we have

$$\mathbb{E}_\beta \mathbb{E}_{(s,a,\tilde{s}',s')}\left[\left\|\tilde{s}' + \beta\psi(s,a)^\top \zeta(\tilde{s}',\beta) - b\right\|^2\right] = \mathbb{E}_\beta \mathbb{E}_{(s,a,\tilde{s}')}\left[\left\|\tilde{s}' + \beta\psi(s,a)^\top \zeta(\tilde{s}',\beta) - b\right\|^2\right].$$

We conclude that

$$\mathbb{E}_\beta \mathbb{E}_{(s,a,\tilde{s}',s')}\left[\left\|\tilde{s}' + \beta\psi(s,a)^\top \zeta(\tilde{s}',\beta) - \sqrt{1-\beta}s'\right\|^2\right]$$

$$= \mathbb{E}_\beta \mathbb{E}_{(s,a,\tilde{s}')}\left[\left\|\tilde{s}' + \beta\psi(s,a)^\top \zeta(\tilde{s}',\beta) - b\right\|^2\right] + \texttt{constant}.$$

Therefore, we obtain the claim. $\qquad\square$

# B Generalization for Partially Observable RL

In this section we discuss how to generalize Diff-SR to Partially Observable MDP (POMDP). We follow the definition of a POMDP given in [Efroni et al., 2022], which is formally denoted as a tuple $\mathcal{P} = (\mathcal{S}, \mathcal{A}, \mathcal{O}, r, H, \rho_0, \mathbb{P}, \mathbb{O})$, where $\mathcal{S}$ is the state space, $\mathcal{A}$ is the action space, and $\mathcal{O}$ is the observation space. The positive integer $H$ denotes the horizon length, $\rho_0$ is the initial state distribution, $r : \mathcal{O} \times \mathcal{A} \to [0,1]$ is the reward function, $\mathbb{P}(\cdot|s,a) : \mathcal{S} \times \mathcal{A} \to \Delta(\mathcal{S})$ is the transition kernel capturing dynamics over latent states, and $\mathbb{O}(\cdot|s) : \mathcal{S} \to \Delta(\mathcal{O})$ is the emission kernel, which induces an observation from a given state.

The agent starts at a state $s_0$ drawn from $\rho_0(s)$. At each step $h$, the agent selects an action $a$ from $\mathcal{A}$. This leads to the generation of a new state $s_{h+1}$ following the distribution $\mathbb{P}(\cdot|s_h, a_h)$, from which the agent observes $o_{h+1}$ according to $\mathbb{O}(\cdot|s_{h+1})$. The agent also receives a reward $r(o_{h+1}, a_{h+1})$. Observing $o$ instead of the true state $s$ leads to a non-Markovian transition between observations, which means we need to consider policies $\pi_h : \mathcal{O} \times (\mathcal{A} \times \mathcal{O})^h \to \Delta(\mathcal{A})$ that depend on the entire history, denoted by $\tau_h = \{o_0, a_0, \cdots, o_h\}$. Let $[H] := \{0, \ldots, H\}$.

The value functions are defined by

$$V_h^\pi(b_h) = \mathbb{E}\left[\sum_{t=h}^H r(o_t, a_t)|b_h\right], \quad Q_h^\pi(b_h, a_h) = r(o_h, a_h) + \mathbb{E}_{\mathbb{P}_b}\left[V_{h+1}^\pi(b_{h+1})\right]. \qquad (24)$$

**Definition 1** (*L-decodability* [Efroni et al., 2022]). $\forall h \in [H]$, *define*
$$x_h \in \mathcal{X} := (\mathcal{O} \times \mathcal{A})^{L-1} \times \mathcal{O},$$
$$x_h = (o_{h-L+1}, a_{h-L+1}, \cdots, o_h). \qquad (25)$$
*A POMDP is L-decodable if there exists a decoder* $p^* : \mathcal{X} \to \Delta(\mathcal{S})$ *such that* $p^*(x_h) = b(\tau_h)$.

That is, under $L$-decodability assumption, it is sufficient to recover the belief state by an $L$-step memory $x_h$ rather than the entire history $\tau_h$. This implies that we can parameterize the Q-value as a function of the observation history $Q_h^\pi(x_h, a_h)$, rather that the unknown belief state $Q_h^\pi(b_h(x_h), a_h)$. By exploiting this $L$-decodability, Zhang et al. [2023a] propose a probable efficient linear function approximation of $Q_h^\pi(x_h, a_h)$ by considering a $L$-step prediction $\mathbb{P}^\pi(x_{h+L}|x_h, a_h)$.

Inspired by this, we apply EBM for $\mathbb{P}^\pi(x_{h+L}|x_h, a_h)$,

$$\mathbb{P}^\pi(x_{h+L}|x_h, a_h) = \exp\left(\psi(x_h, a_h)^\top \nu(x_{h+L}) - \log Z(x_h, a_h)\right), \tag{26}$$

$$Z(x_h, a_h) = \int \exp\left(\psi(x_h, a_h)^\top \nu(x_{h+L})\right) \mathrm{d}\, x_{h+L}. \tag{27}$$

We then apply the techniques presented in Section 3.2 to learn diffusion representation $\phi(x_h, a_h) \in \mathbb{R}^d$ as an approximation to the random features

$$\phi_\omega(x_h, a_h) = \exp\left(-\mathbf{i}\omega^\top \psi(x_h, a_h)\right) \exp\left(\|\psi(x_h, a_h)\|^2/2 - \log Z(x_h, a_h)\right). \tag{28}$$

The learned representation is subsequently utilized to parameterize the value function $Q_h^\pi(x_h, a_h)$ for policy optimization as demonstrated by [Zhang et al., 2023a]. In particular, we consider value function approximation $Q_{\xi,\theta}(x_h, a_h) = \phi_\theta(x_h, a_h)^\top \xi$ and update it by

$$\ell_{\mathrm{critic}}(\xi) = \mathbb{E}_{x,a,r,x'\sim\mathcal{D}}\left[\left(r + \gamma\mathbb{E}_{a'\sim\pi}[Q_{\bar\xi,\theta}(x', a')] - Q_{\xi,\theta}(x, a)\right)^2\right], \tag{29}$$

The policy, now conditioned on the $L$-step history $x$, is updated by $\max_\pi \mathbb{E}_{x\sim\mathcal{D},a\sim\pi}[\min_{i\in\{1,2\}} Q_{\xi_i,\theta}(x, a)]$. We refer interested readers to [Zhang et al., 2023a] for more details on representation learning in POMDPs.

## C   Details and Analysis for Fully Observable MDP Experiments

Table 2: Hyperparameters used for Diff-SR in state-based MDP environments.

| Hyperparameter | Value |
| --- | --- |
| Actor Learning Rate | 0.003 |
| Critic Learning Rate | 0.0003 |
| Learning Rate for $\psi, \zeta, \theta$ | 0.0001 |
| Actor Hidden Layer Dimensions | (256, 256) |
| Diff-SR Representation Dimension | 256 |
| Discount factor $\gamma$ | 0.99 |
| Critic Soft Update Factor $\tau$ | 0.005 |
| Batch Size | 1024 |
| Number of Noise Levels | 1000 |
| $\psi$ Network Width | 256 |
| $\psi$ Network Hidden Depth | 1 |
| $\zeta$ Network Width | 512 |
| $\zeta$ Network Hidden Depth | 1 |

### C.1   Baseline Methods

For baseline methods, we include ME-TRPO [Kurutach et al., 2018], PETS [Chua et al., 2018], and the best model-based results among Luo et al. [2021], Deisenroth and Rasmussen [2011], Heess et al. [2015], Clavera et al. [2018], Nagabandi et al. [2018], Tassa et al. [2012] and Levine and Abbeel [2014] from MBBL [Wang et al., 2019]. For model-free algorithms, we include PPO [Schulman et al., 2017], TRPO [Schulman et al., 2015] and SAC [Haarnoja et al., 2018].

### C.2   Experiment Setups

We implemented our algorithm based on Soft Actor-Critic (SAC). We use *feature update ratio* to denote the frequency of updating the diffusion representations as compared to critic updates. We

Table 3: Performance on various continuous control problems with partial observation. We average results across 4 random seeds and a window size of 10K. Diff-SR achieves a similar or better performance compared to the baselines. Here, Best-FO denotes the performance of Diff-SR using full observations as inputs, providing a reference on how well an algorithm can achieve most in our tests.

|  | HalfCheetah | Humanoid | Walker | Ant | Hopper | Pendulum |
|---|---|---|---|---|---|---|
| **Diff-SR** | **3864.2 ± 482.3** | 650.1 ± 57.4 | **1860.5 ± 912.1** | **3189.7 ± 720.7** | **1357.6 ± 506.4** | **167.4 ± 4.4** |
| $\mu$LV-Rep | 3596.2 ± 874.5 | **806.7 ± 120.7** | 1298.1 ± 276.3 | 1621.4 ± 472.3 | 1096.4 ± 130.4 | 168.2 ± 5.3 |
| Dreamer-v2 | 2863.8 ± 386 | 672.5 ± 36.6 | 1305.8 ± 234.2 | 1252.1 ± 284.2 | 758.3 ± 115.8 | **172.3 ± 8.0** |
| SAC-MLP | 1612.0 ± 223 | 242.1 ± 43.6 | 736.5 ± 65.6 | 1612.0 ± 223 | 614.15 ± 67.6 | 163.6 ± 9.3 |
| SLAC | 3012.4 ± 724.6 | 387.4 ± 69.2 | 536.5 ± 123.2 | 1134.8 ± 326.2 | 739.3 ± 98.2 | 167.3 ± 11.2 |
| PSR | 2679.75 ±386 | 534.4 ± 36.6 | 862.4 ± 355.3 | 1128.3 ± 166.6 | 818.8 ± 87.2 | 159.4 ± 9.2 |
| PolyGRAD | 987.1 ± 374.6 | 764.5 ± 91.2 | 211.4 ± 100.4 | 493.0 ± 95.4 | 527.8 ± 97.7 | 174.0 ± 6.1 |
| Best-FO | 7223.5±437.15 | 823.1±118.9 | 3722.2±406.1 | 4788.6±623.1 | 2800.5±95.4 | 162.3±3.0 |

sweep the value of this parameter within [1, 3, 5, 10, 20] for all MuJoCo experiments and report the configuration that achieved the best results. Other hyper-parameters are listed in Table 2. For evaluation, we test all methods every 5,000 environment steps by simulating 10 episodes and recording the cumulative return. The reported results are the average return over the last four evaluations and four random seeds.

## D  Computational Efficiency and Runtime Comparison

Full results are presented in Figure 4. We use exactly the same experimental setups as used in Table 1, which has been described in 5.1. We observe that in all cases, Diff-SR is about $4\times$ faster than PolyGRAD. Such efficiency can be attributed to the fact that while we utilize diffusion to train the representations, we do not have to sample from the diffusion model iteratively, which is the main computational bottleneck for SOTA diffusion-based RL algorithms.

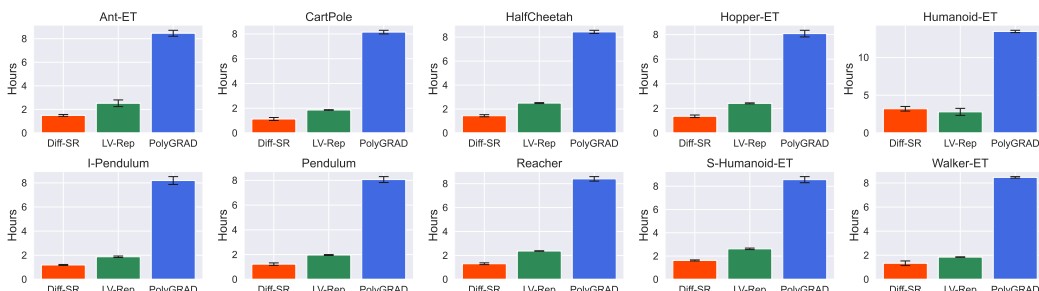

Figure 4: Per-task runtime of Diff-SR, LV-Rep and PolyGRAD on tasks from MBBL.

## E  Details and Analysis for State-based Partially Observable MDP Experiments

### E.1  Implementation Details

We also experiment Diff-SR with state-based Partially Observable MDP (POMDP) tasks. We contruct a partially observable variant based on OpenAI gym MuJoco [Todorov et al., 2012]. Adhereing to standard methodology, we mask the velocity components within the observations presented to the agent, effectively rendering the tasks partially observable [Ni et al., 2021, Weigand et al., 2021, Gangwani et al., 2020]. Under this masking scheme, a single observation is insufficient for decision-making. Therefore, the agent must aggregate past observations to infer the missing information and select appropriate actions. The reconstruction of missing information can be done by aggregating a past history of length $L$, as demonstrated in Definition 1.

The POMDP experiment follows a similar setup to the fully observable one described in Appendix C. In the partially observable setting, velocity information is masked, and we concatenate the past $L = 3$ observations follow [Zhang et al., 2023a]. The universal hyperparameters used for POMDP are listed in Table 4. For each individual environment, we explored an array of parameters and chose the highest-performing configuration. Specifically, the critic and actor learning rates were varied across $\{0.0015, 0.00015\}$, the model learning rate was varied across $\{0.0001, 0.0003, 0.0008\}$ and the feature update ratio was varied across $\{1, 3, 5, 10, 20\}$.

We evaluate six baselines in our experiments: a diffusion approach, PolyGRAD [Rigter et al., 2023], two model-based approaches, Dreamer [Hafner et al., 2019a, 2021] and Stochastic Latent Actor-Critic (SLAC) [Lee et al., 2020], a model-free baseline, SAC-MLP, that concatenates history sequences (past four observations) as input to an MLP layer for both the critic and policy, and the neural PSR [Guo et al., 2018]. We also compared to a representation-based baseline, $\mu$LV-Rep [Zhang et al., 2023a]. All methods are evaluated using the same procedure as in the fully observable setting. As a reference, we also provide the best performance achieved in the fully observable setting (without velocity masking), denoted as Best-FO, which serves as a benchmark for the optimal result an algorithm can achieve in our tests.

## E.2   Results and Analysis

Table 3 presents all experiment results, showing effectiveness of Diff-SR in partially observable continuous control tasks. The proposed method delivers superior results in 4 out of 6 tasks (HalfCheetah, Walker, Ant, Hopper). It significantly outperforms other algorithm in Walker and Ant, and achieved a comparable result with the lowest standard deviation on Pendulum, indicating consistent performance.

The wall time comparison between Diff-SR and PolyGRAD is shown in Figure 5. In the Humanoid task, Diff-SR is approximately 3 times faster than PolyGRAD, whereas in the other tasks, it is about 4 times faster.

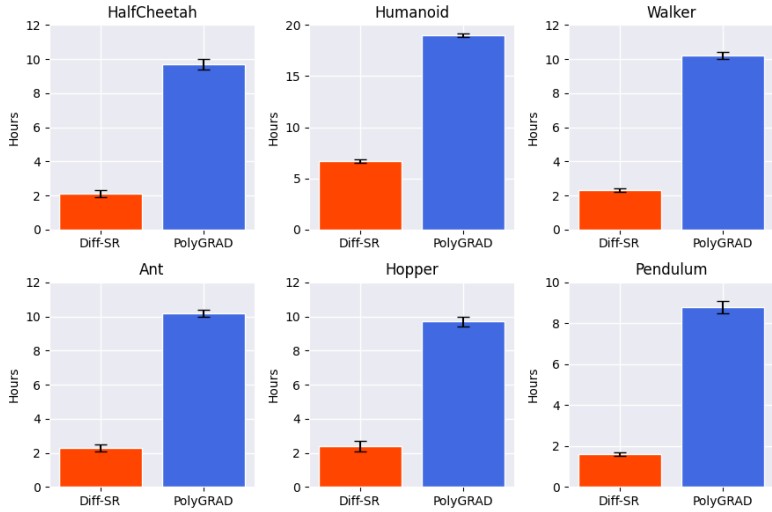

Figure 5: Per-task running time of Diff-SR and PolyGRAD on tasks from MBBL with partial observation.

## E.3   Ablations And Modifications

**Masking Observations**   Observations consist of both velocities and positions. Masking velocities, rather than positions, more accurately reflects real-world scenarios where positions are typically directly observed, whereas velocities must be inferred. In the humanoid environment, we specifically mask only the q-velocity component.

Table 4: Hyperparameters used for Diff-SR in state-based POMDP experiments.

| Hyperparameter | Value |
| --- | --- |
| Actor Hidden Layer Dimensions | (512, 512) |
| Diff-SR Representation Dimension | 512 / 128 (cheetah) |
| Discount Factor $\gamma$ | 0.99 |
| Critic Soft Update Factor $\tau$ | 0.005 |
| Batch Size | 1024 |
| Number of Noise Levels | 1000 |
| $\psi$ Network Width | 512 |
| $\psi$ Network Hidden Depth | 1 |
| $\zeta$ Network Width | 256 |
| $\zeta$ Network Hidden Depth | 1 |

**Random Action** At the beginning of training and evaluation, we randomly sample actions from the action space. This is necessary because we use concatenated observations to sample actions from the policy. This approach might explain the larger standard deviation observed in POMDP compared to the fully observable case.

# F  Image-based Partially Observable MDP Experiment

## F.1  Implementation Details

Instead of directly diffusing over the raw pixel space, we used the VAE structure from Zhang et al. [2023a] to first encode the pixel observation $o$ into a 1-D latent embedding $e$. To deal with the partial observability, we also follow Zhang et al. [2023a] to concatenate the embeddings of the past three observations (denoted as $o^3$) together as the state of the agent, denoted as $e^3$. Let the next frame be $o'$ and its embedding be $e'$, the representation learning objective thus translates to fitting the score function $\psi(e^3, a)$ and $\zeta(\tilde{e}', \beta)$, i.e. performing the diffusion in the latent space. In the following paragraphs, we will detail the architectures for Diff-SR.

**Diffusion Representation Learning.** Unlike previous state-based experiments, we formalize the network $\psi$ and $\zeta$ using the LN_ResNet architecture proposed by IDQL [Hansen-Estruch et al., 2023]. Compared to standard MLP networks, LN_ResNet is equipped with layer normalization and skip connections, making it expressive enough for diffusion modeling. For representation learning, it is worthwhile to note that, apart from the loss objectives defined in Eq (18), we also preserve the gradients of the representation networks and train them with the critic's loss defined in (21). This design choice aligns with previous works [Zhang et al., 2023a] and encourages the representations to contain task-relevant information.

**RL optimization.** We develop our code based on DrQ-V2 [Yarats et al., 2021], a model-free RL algorithm designed for tasks with visual inputs. We preserved most of the design choices from DrQ-V2, except for the architectures of the actor and the critic. For the actor network, it receives the concatenated embeddings $e^3$ and outputs an action. The critic networks receive the diffusion representation $\phi_\theta$ as defined in the main text and predict the Q-values.

The hyper-parameters for the score functions $\psi, \zeta$, the actor and the critic networks are listed in Table 5.

## F.2  Generation Results

In addition to the learning curves, we also present the qualitative generation results using the learned score functions. Figure 6 illustrates the progression of the diffusion process over time. In the figure, we sample a batch of data $(o^3, a, o')$ from the replay buffer, and the first row depicts the ground-truth target image $o'$. Starting from the second row, we sample a random Gaussian noise $\tilde{e}'_{1000} \sim \mathcal{N}(0, I)$ and iteratively apply the learned reverse diffusion process using the guidance of $(e^3, a)$ to generate the latent embeddings at various stages of the diffusion. For every 200 steps, we pass the latent embedding to the decoder to obtain reconstructed images $\tilde{o}'_t$ and visualize them in the figure.

Table 5: Hyperparameters used for Diff-SR in image-based POMDP experiments.

| Hyper-parameters | Value |
|---|---|
| Actor Learning Rate | 0.0001 |
| Critic Learning Rate | 0.0001 |
| Learning Rate for $\psi, \zeta, \theta$ | 0.0003 |
| Actor Hidden Layer Dimensions | (1024, 1024) |
| Diff-SR Representation Dimension | 1024 |
| Discount Factor $\gamma$ | 0.99 |
| Critic Soft Update Factor $\tau$ | 0.01 |
| Batch Size | 1024 |
| Number of Noise Levels | 1000 |
| LN_ResNet Layer Width for $\psi$ | 512 |
| LN_ResNet Layer Width for $\zeta$ | 512 |
| # of LN_ResNet Layers for $\psi$ | 4 |
| # of LN_ResNet Layers for $\zeta$ | 2 |

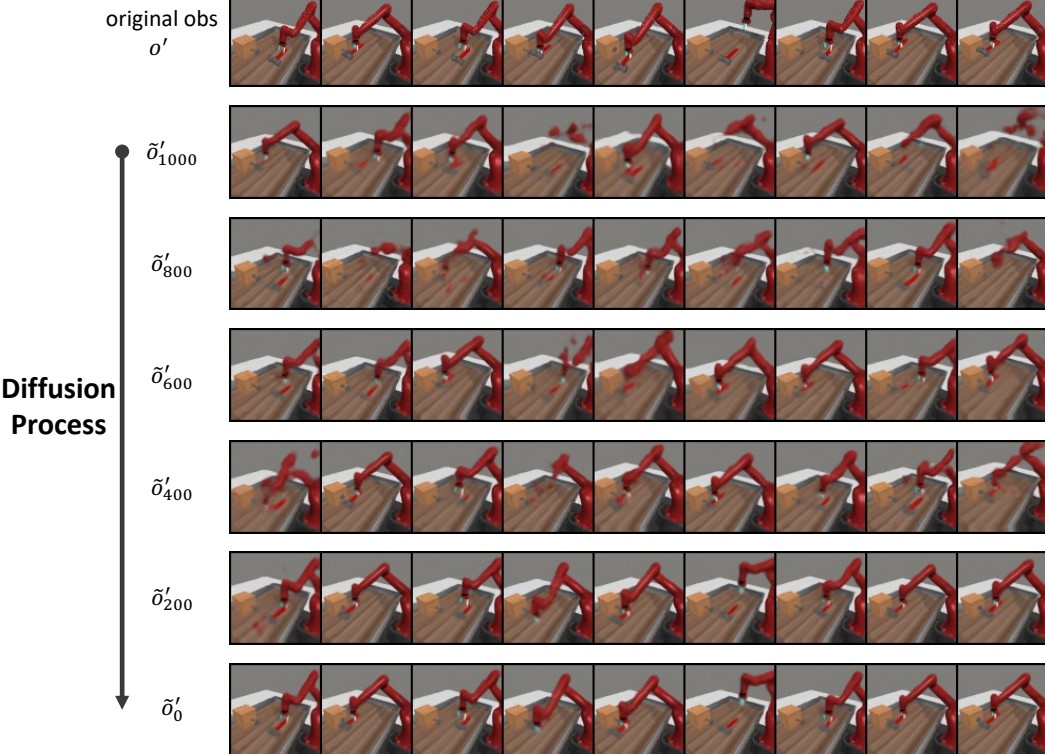

Figure 6: The generation results of Diff-SR.

The second row corresponds to the initial noisy embeddings and exhibits significant distortion and noise. As the denoising steps progress, we observe a gradual reduction in noise and an increasing clarity in the generated images. By the time we reach $\tilde{o}'_{600}$, the overall structure of the scene becomes more discernible, although some artifacts remain. Further along the denoising process, the images exhibit substantial improvements in terms of detail and realism. The final output, $\tilde{o}'_0$, closely resembles the original observations, indicating the effectiveness of our score functions in capturing the underlying information about the data.

