# OpenReview forum: "Diffusion Spectral Representation for Reinforcement Learning"
_NeurIPS.cc/2024/Conference — NeurIPS 2024 poster_

### Official Review · Reviewer_iUqH · 2024-06-28

**Soundness:** 3
**Presentation:** 3
**Contribution:** 3
**Rating:** 6
**Confidence:** 3

**Summary:**

This paper proposes an efficient and novel method for integrating diffusion models into reinforcement learning (RL). One major drawback of existing diffusion models is the high computational cost at inference time due to iterative sampling. This paper utilizes a diffusion model to learn representations of latent structures for the transition function. These representations can then be used for the value function in the RL framework. The authors evaluated their method on both MDP and POMDP tasks, showing significant benefits on the POMDP tasks compared to the baselines.

**Strengths:**

The writing is generally good to follow though some editing on the notations could make the reading easier especially in section 3.1 and 3.2. And the author did a good job on motivating from LV-Rep.

The proposed method, which integrates the diffusion model into the LV-Rep framework, is novel and offers a different perspective on existing diffusion-based RL methods. This method outperforms current diffusion-based RL techniques on POMDP tasks while consuming less runtime.

**Weaknesses:**

In Table 1, for the MDP setting, Diff-Rep only outperforms other relevant baselines in the HalfCheetah task while showing similar or worse performance compared to LV-Rep and PolyGRAD. It might be beneficial to emphasize the advantages of the proposed algorithm in the POMDP setting.  In addition, an explanation for this lower performance would be necessary instead of claiming "significantly better of comparable performance to all baseline methods in most tasks except for Humanoid-ET".

While the runtime of the proposed method is lower than other diffusion-based methods, Diff-Rep loses the flexibility of conditioning the policy at inference time.

Minor
Some derivations/explanations are needed for the approximation in Eq 12.

**Questions:**

Could the author provide a runtime comparison between Diff-Rep and LV-Rep?

Could the author provide some plots showing the accuracy of the estimated Q function when using the proposed representation?

Could the author suggest any methods that might help interpret the latent representation?

---

> ### Author Rebuttal · Authors · 2024-08-07
>
> * **W1: Table 1 performance**
>
> We would like to note that our method surpasses or parallels existing methods for 7 out of 10 tasks from the MBBL benchmark. We keep the same architecture and hyper-parameters when benchmarking our approach, so we expect better performances for the rest of the tasks if the architecture can be tuned specifically for them.
>
>
> * **W2: Diff-Rep lacks the flexibility of conditioning the policy at inference time?**
>
> We acknowledge that the current version of Diff-Rep lacks the flexibility of goal-conditional generation at inference time since we are modeling the transition model to handle common RL problems. However, our approach preserves the capability of conditional generation by using the goal information to condition the scoring network $\zeta$. We believe this presents an intriguing direction for future research.
>
>
> *  **W3: Some minor derivations or explanations are needed for the approximation in Equation 12**
>
> In Equation 12, we parameterize the perturbed transition operator $\mathbb{P}(\tilde{s}'|s, a; \beta)$ as an EBM model, whose energy function $\psi(s, a)^\top \nu(\tilde{s}', \beta)$ is factorized by two representation functions $\psi$ and $\nu$. As noted in line 213 as well as in [1], such parameterization possesses the universal approximation capability as long as $\psi$ and $\nu$ are capable of universal approximation. Therefore, the parameterization in Equation 12 is sound since we are using neural networks for both $\psi$ and $\nu$.
>
> [1] Hiroaki Sasaki and Aapo Hyvärinen. Neural-kernelized conditional density estimation. 2018.
>
>
>  * **Q1: Runtime comparison between Diff-Rep and LV-Rep?**
>
> The runtime comparison is illustrated in the attached Figure of the general rebuttal. Note that Diff-Rep is more computationally efficient than both LV-Rep and PolyGRAD in most of the tasks from MBBL.
>
>
> * **Q2: showing the accuracy of the estimated Q function when using Diff-Rep?**
>
> Illustrating the accuracy of the estimated $Q$ functions in control tasks is generally challenging, as obtaining ground truth values necessitates Monte Carlo estimation, which suffers from great variance as the task horizon increases. Therefore, we conducted a toy experiment in a grid world environment, where the ground truth $Q$-values can be solved accurately via policy evaluation. Please refer to the general response for details.
>
> * **Q3: Suggest some methods to help interpret the latent representation?**
>
> Diff-Rep extracts $\psi(s, a)$ which captures the transition functions and constructs sufficient representations that can represent the $Q$-functions of any policy $\pi$ based on our theory. Thus, one way to interpret the quality of the representation is by inspecting the generation results to see whether $\psi$ faithfully captures the dynamics information. We included the generation results in Section F.2.

---

> > ### Comment · Reviewer_iUqH · 2024-08-12
> >
> > Thank you for the reply.  I will keep my current score.

---

### Official Review · Reviewer_sdAM · 2024-07-02

**Soundness:** 2
**Presentation:** 2
**Contribution:** 3
**Rating:** 6
**Confidence:** 3

**Summary:**

## Main summary

The authors propose Diff-Rep, an algorithmic framework leveraging diffusion models for learning spectral representations of MDPs, from which relevant quantities such as the Q-function can be linearly decoded.

The main methodological contribution of the paper is showing that spectral representations can be learned using a diffusion-like loss function, obtained by exploring the connection between energy-based models (EBMs) and diffusion models. The resulting method, however, does not require sampling from the diffusion model, and so requires less wall time. The authors also show how to perform online policy optimization using Diff-Rep.

They evaluate their method on Gym-MuJoCo locomotion tasks, which are MDP-based, and on image-based partially observable MDP tasks. Diff-Rep achieves similar or superior performance to baseline methods on most locomotion tasks, and consistently surpasses the performance and wall-time of  PolyGRAD, a recent diffusion-based trajectory prediction model.

## More detailed summary of methods

Their framework assumes the transition function $\mathbb{P}(s’ | s, a)$ can be written as an inner product of state-action features $\phi^*(s, a)$ and next-state features $\mu(s’)$: $\mathbb{P}(s’ | s, a) = \langle \phi^*(s, a), \mu(s’) \rangle$.

The authors propose modeling this transition function using an Energy-Based Model (EBM), from which they obtain explicit forms for $\phi^*$ and $\mu^*$ as random Fourier features.

Next, by considering a noised next state $\tilde{s}’$, they obtain a relation that must be satisfied by the energy term of this EBM. They then derive from this relation a diffusion-like loss which can be used to train the parameters of the EBM.

To learn the state-action features $\phi^*$, they further include a regularization term incentivizing orthonormality of the entries of $\phi^*$. $\phi^*$ can then be optimized jointly with the EBM, sharing many similarities with the training of Denoising Diffusion Probabilistic Models.

Finally, the authors show how to jointly learn representations $\phi^*$, a Q-function, and a policy $\pi$ online, leveraging $\phi^*$ to parameterize the Q-function. They further propose including an Upper Confidence Bound (UCB) bonus to rewards during optimization to incentivize exploration.

**Strengths:**

1. Significance of research question: leveraging the flexibility and representation power of diffusion models has been actively researched recently, and their expensive sampling loop has proved a challenge (e.g. Janner et al. 2022, section 5.4). The authors consider the problem of reaping the benefits of diffusion in decision-making, but without incurring in this inefficiency.


1. Originality: while learning spectral representations for MDPs was already present in prior work, the authors’ approach of parameterizing the transition function and spectral representations using an EBM and training it via a diffusion-like loss function is original.

1. Quality of exposition and overall quality: I believe these can be significantly improved. See Weaknesses.

1. Quality of evaluation: the authors compare the performance of Diff-Rep on Gym-MuJoCo locomotion tasks (e.g. Hopper, HalfCheetah) and on a partially-observable image-based task from the Meta-World benchmark. They consider several model-based and model-free baselines, as well as other techniques based on learning representations. They demonstrate performance gains on most MDP and POMDP tasks, and also highlight a gain in wall time compared to PolyGRAD (Rigter et al. 2024).

1. In addition, they give qualitative results showing the diffusion model trained via Diff-Rep is able to reconstruct scenes with good fidelity in the setting of their POMDP experiment.

**Weaknesses:**

1. Is the singular value decomposition of the transition operator taken as an assumption, or do the authors claim it holds for any MDP? Other works (e.g. Ren et al. 2022a) list this decomposition as an assumption, but the phrasing in line 82 (“one can always factorize [...]”) seems to claim that it holds in general, provided one does not require the representations to take values in finite dimensions. If the authors do claim this, I believe either a citation or a proof would be warranted.

1. On a similar note, the authors claim on line 63 that they show “diffusion-based representations are sufficiently expressive to represent the value function of any policy”. It is not clear to me that either the exposition of the methods or the experiments prove this in such generality.

1. The claim on line 93 that “the learning of the ϕ* is essentially equivalent to un-normalized conditional density estimation” seems unsupported by a citation or an explanation. Similarly for the claim that “learning requires full-coverage data, while the exploration strategy requires an accurate ϕ*” in line 96.

1. Considering the central role of EBMs and random Fourier features in the derivation of the method, I believe a brief exposition about them in Section 2 would be warranted, in particular to clarify the $\langle \cdot, \cdot, \rangle_{\mathcal{N}(\omega)}$ notation in Equation 9.

1. Further, appropriate citations for EBMs seem to be missing in Section 3.2.

1. An explicit definition of $\nu(\tilde{s}’, \beta)$ is missing in Equation 12.

Clarity and writing:

The paper contains a relatively large number of what seem to me like grammar mistakes. Here is a non-exhaustive list of examples and suggestions for edits:

1. Line 29: “Exploring the potential of diffusion models for sequential decision-making is increasingly being investigated.” -> “The potential of diffusion models for sequential decision-making is increasingly being investigated.”

1. Line 50: “to avoid suboptimal policy” -> “to avoid a suboptimal policy”

1. Line 99: “the existing methods either require [...]  or relies on” -> “the existing methods either require [...]  or rely on”

1. Line 105: “After T-step” -> “After T steps”

1. Line 110: “Recognize the diffusion models are [...], the ELBO naturally as a choice for learning,” -> “Recognizing that diffusion models are [...], maximizing the ELBO arises as a natural choice:”

1. Line 117: “as a neural networks” -> “as a neural network”

1. Line 118: “the samples can be generated from [...], and then, following [...]” -> “the samples can be generated by sampling [...], and then following [...]”

1. Line 155: “We apply the EBM for transition operator” -> “We parameterize the transition operator using an EBM”

1. Line 176: “we consider the samples from dynamics s ′ ∼ P(s ′|s, a) is perturbed with Gaussian noise” -> “we consider perturbing the samples from dynamics s ′ ∼ P(s ′|s, a) with Gaussian noise”

1. Line 178: “$\mathbb{P}(\tilde{s}’|s, a; \alpha) \to \mathbb{P}(s’|s, a)$” -> “$\mathbb{P}(\tilde{s}’|s, a; \beta) \to \mathbb{P}(s’|s, a)$”

**Questions:**

1. In line 177, why is noising chosen to be $\mathbb{P}(s’|s, a) = \mathcal{N}(\sqrt{1-\beta}s’, \beta \sigma^2 I)$, as opposed to $\mathbb{P}(s’|s, a) = \mathcal{N}(\sqrt{1-\beta}s’, \beta I)$, as is normally done, e.g. in Section 2? The presence of the $\sigma^2$ factor means the variance need not be preserved. Is this intentional? If so, what drove this design decision, and how do you choose $\sigma^2$?

1. In line 591, it seems to me that the second term equals 0 by applying the tower property conditioning on $\tilde{s}’$, $s$ and $a$. Is this correct? It might be good to make this explicit.

**Limitations:**

The authors include one sentence addressing limitations in the conclusion, referencing the fact that their method has not been evaluated on real-world data.

My view is that the main limitation of this work is exposition and clarity, which would warrant significant revision before publication. Particularly critical are unjustified claims, lack of clarity regarding the assumptions of the method, and several grammar/writing issues, as outlined in the Weaknesses section.

---

> ### Author Rebuttal · Authors · 2024-08-07
>
> We apologize for any ambiguity regarding certain claims in the paper. Below, we will offer additional justifications for them and revise the corresponding sections. We also appreciate the grammar corrections and will ensure that the final version is free of such errors.
>
> * **W1: SVD decomposition of transition operator assumption**
>
> The SVD of the transition operator $T(s' | s, a)$ holds in general MDP settings, but potentially with infinite-dimensional spectrum.
> When the spaces $\mathcal{S}$ and $\mathcal{A}$ are finite, $T$ can be represented as a matrix $T\in \mathbb{R}^{(|\mathcal{S}|\cdot |\mathcal{A}|)\times |\mathcal{S}|}$, which admits a finite SVD.
> For continuous $\mathcal{S}$ and $\mathcal{A}$, if $T(s'|s, a)$ can be represented by a discrete latent variable model, i.e.,
>
> $$T(s'|s, a) = \sum_{i=1}^k p(s'|i)p(i|s, a),$$
>
> we obtain a low-rank spectral decomposition as $\phi^*(s, a)=[p(i|s, a)]_{i=1}^k$ and $\mu^*(s')=[p(s'|i)]\_{i=1}^k$. Generally, in cases where these spaces are infinite (for example, in continuous settings), a countably infinite SVD of the transition operator $T$ exists, provided that $T$ is compact. For additional information, please refer to [1].
>
> [1] Jordan Bell. The singular value decomposition of compact operators on Hilbert spaces. 2014.
>
> * **W2: Why Diff-Rep is sufficiently expressive to represent the value function of any policy.**
>
> From line 82 to line 91, we show that with the SVD of the transition operator $\mathbb{P}(s'|s, a)=\langle\phi^*(s, a), \mu^*(s, a)\rangle$, we can represent $Q^\pi(s, a)$ as a linear function w.r.t. $r(s, a)$ and $\phi^*(s, a)$. Our paper derives such representations via diffusion (Section 3.2), thus enjoying the same property. We also provide a toy experiment in the general response to examine this property.
>
> * **W3: Claims about $\phi^{*}$**
>
> **Un-normalized conditional density estimation**:
>
> As $\phi^*(s, a)$ comes from the SVD of the conditional density $\mathbb{P}(s'|s, a)$ (Equation 1), learning such representation translates into estimating the conditional density $\mathbb{P}(s'|s, a)$ using available samples. However, vanilla parameterization of $\phi^*$ and $\mu^*$ cannot guarantee the normalizing condition of probability distributions, i.e. $\int \mathbb{P}(s'|s, a)\mathrm{d}s'=1$, thus we refer to it as un-normalized density estimation.
>
> **Learning requires full-coverage data**:
>
> Learning an accurate $\phi^*$ requires data with full coverage to capture the underlying conditional density $\mathbb{P}(s' |s, a)$.
>
> **Exploration requires accurate $\phi^{*}$**:
>
> In the online setting, the data is progressively collected by the agent. To encourage the exploration of unvisited regions, existing approaches design exploration strategies that rely on $\phi^*$ (e.g., [1-2]). This dilemma has sparked a series of provable and practical RL algorithms [3-4], including our method.
> We will include relevant citations in the revision.
>
> [1] Chi Jin, et al. Provably efficient reinforcement learning with linear function approximation. 2020.
>
> [2] Zhuoran Yang, et al. Provably efficient reinforcement learning with kernel and neural function approximations. 2020.
>
> [3] Alekh Agarwal, et al. Flambe: Structural complexity and representation learning of low rank mdps. 2020.
>
> [4] Tongzheng Ren, et al. Latent variable representation for reinforcement learning. 2022.
>
> * **W4: Random Fourier features and explanations.**
>
> Thanks for pointing this out. Random Fourier feature [1] is an important technique to recover the SVD from EBM parameterization. We will provide a more detailed exposition of this in Section 2.
>
> Specifically, $\langle \cdot, \cdot\rangle_{\mathcal{N}(\omega)}$ denotes the dot product under the distribution $\mathcal{N}(\omega)$. Equation 9 is obtained by a direct application of the random Fourier feature to Gaussian kernels:
> \begin{align}
>     \exp(-\frac{\\|x-y\\|^2}{2}) &=\int p(\omega)\exp(-\mathbf{i}\omega^\top(x-y))\mathrm{d}\omega \\\\
>        &=\int p(\omega)\exp(-\mathbf{i}\omega^\top x)\exp(\mathbf{i}\omega^\top y)\mathrm{d}\omega \\\\
>        &=\mathbb{E}_{\omega\sim p(\omega)} \left[\exp(-\mathbf{i}\omega^\top x)\exp(\mathbf{i}\omega^\top y)\right] \\\\
> \end{align}
>
> where $p(\omega)\sim\mathcal{N}(0, I_d)$.
>
> [1] Ali Rahimi, Benjamin Recht, et al. Random features for large-scale kernel machines. NIPS. 2007.
>
> * **W5: Citations for EBMs**
>
> Thanks for pointing this out. There is a substantial amount of literature related to EBMs, and we have already cited some seminal works. We will include additional references in Section 3.2.
>
> * **W6: An explicit definition of $\nu$**
>
> In Equation 12, we parameterize the perturbed transition operator $\mathbb{P}(\tilde{s}'|s, a; \beta)$ as an EBM, whose energy function $\psi(s, a)^\top \nu(\tilde{s}', \beta)$ is composed by two representation functions $\psi$ and $\nu$. The representation function $\nu(\tilde{s}', \beta)$ receives the corrupted state $\tilde{s}'$ and the noise level $\beta$ as input, and outputs a possibly infinite-dimensional vector. We will update the paper to clarify this notation.
>
> * **Q1: The presence of $\sigma^2$ is unnecessary.**
>
> We apologize for the confusing expressions. The variance $\sigma^2$ should be removed and the corruption is defined by $\beta$ entirely. We will revise to remove $\sigma^2$ throughout the paper.
>
> * **Q2: Why does the second term equal 0 in line 591.**
>
> This is a direct application of the tower property. Specifically,
>
> \begin{align}
> &\mathbb{E}\_\beta \mathbb{E}\_{(s, a, \tilde{s}', s')} [(\tilde{s}'+\beta\psi(s, a)^\top\zeta(\tilde{s}',\beta)-b)^\top (b-\sqrt{1-\beta}s')] \\\\
> &=\mathbb{E}\_\beta \mathbb{E}\_{s, a, \tilde{s}'}[(\tilde{s}'+\beta\psi(s, a)^\top\zeta(\tilde{s}',\beta)-b)^\top (b-\sqrt{1-\beta}\mathbb{E}_{\mathbb{P}(s'|\tilde{s}', s, a, \beta)}[s'])] \\\\
> &=0.
> \end{align}
>
> where the first equation is due to the independence between $b$ and $s'$; the second equation comes from the definition of $b$.

---

> > ### Comment · Reviewer_sdAM · 2024-08-10
> > **Response to rebuttal**
> >
> > I thank the reviewers for their careful response to my reported concerns. A few follow-up questions:
> >
> > # W1
> > - What do the authors mean by spectrum of T in the fully general case with no assumptions on S and A?
> > - What do the authors mean by an infinite-dimensional spectrum? Do the authors mean instead an infinite spectrum?
> > - My understanding is that the transition operator maps a pair (s, a) to a probability measure over s', which can potentially be represented as a density (i.e. a function of s, a and s') if it is absolutely continuous with respect to some reference measure. Since S and A need not be vector spaces in general, and since the space of probability measures over a set is not a vector space (due to the need that such measures integrate to 1), I don't see how T could be regarded as an operator between e.g. normed vector spaces in the fully general setting. Does the SVD you reference hold when S and A are allowed to be a completely arbitrary set, and T is allowed to be a completely arbitrary kernel mapping S x A to a probability measure over S?
> > - One can consider a "continuous" setting such as the control of a point mass inside a bounded region. In this case, the state space is infinite, but is not a vector space. In what way do you apply results of spectral theory in Hilbert spaces in this situation?
> > - What does it mean for T to be compact in this case (S is "continuous" but not a vector space)? Does it involve something like "mapping bounded sets in S x A to relatively compact sets of probability measures on S"?
> >
> > In general, I believe it would be most efficient to find a sufficiently restricted setting in which you wish to claim the existence of the SVD in a way that remains relevant for this work. The argument for full generality is unfortunately not yet clear to me.
> >
> > # W2
> >
> > It seems to me that whether this argument follows depends on the generality of the existence of the SVD, of which I am not yet convinced. It would be great if the authors could provide further clarification (ideally including specific references to literature on this result).
> >
> > # W3
> >
> > - Un-normalized conditional density estimation: I am satisfied with this explanation.
> >
> > - Learning requires full-coverage data: it seems intuitive to me that data with full coverage would be a sufficient condition for learning the underlying conditional density, but not so much that it is impossible to learn $\mathbb{P}(s' |s, a)$ with acceptable accuracy without full coverage.
> >
> > In general, the notions of full coverage and necessary conditions for "learning $\phi^*$" don't seem to be defined precisely in the context of this statement. Because of this, wording emphasizing the qualitative nature of the statement seems more appropriate (e.g. "learning $\phi^*$ can be challenging without full-coverage data").
> >
> > - Exploration requires accurate $\phi^*$: do the referenced works show that exploration without accurate $\phi^*$ is impossible, or that exporation with $\phi^*$ is possible?
> >
> > # W4, W5, W6, Q1 and Q2
> >
> > Thank you for agreeing to include these in the paper. I believe it will help clarity and accessibility, and make it more self-contained.
> >
> > With the above, I believe most formatting and clarity concerns will be addressed. My remaining queries mostly concern the mathematical and scientific precision of certain statements, but I think these can be easily remedied by simply making the claims in less generality or in a more qualitative way (e.g. "learning might be a challenge in practice" instead of "learning would be impossible in principle").
> >
> > Because of this, I will raise my score.

---

> > > ### Author Response · Authors · 2024-08-12
> > >
> > > Thank you for your appreciation and we are delighted to know that the majority of the concerns have been addressed. Below, we provide explanations for the follow-up questions.
> > >
> > > **W1**
> > > For the first question, we interchangeably use "infinite-dimensional spectrum" and "infinite spectrum" in the response.
> > >
> > > With the compactness of an operator defined as in https://en.wikipedia.org/wiki/Compact_operator. We argue that
> > >
> > > 1),  It is known that the compactness of the operator implies the existence of a countable SVD (potentially infinite-dimension, but still countable);
> > >
> > > 2), We argue that the conditional distribution defined on normed vector space is still an operator, even if it is a distribution;
> > >
> > > 3), In (Ren et al., 2022), there is an additional assumption that the decomposed representations $\phi$ and $\mu$ are also distribution, while we did not make such an assumption.
> > >
> > > Therefore, we have the claim that for arbitrary conditional distribution over *normed vector space* in MDP, we can obtain an SVD, while in (Ren et al., 2022) it becomes an assumption. In fact, for continuous control setting, [1] provides an excellent example of the SVD, where the general stochastic nonlinear control model can be factorized with an infinite-dimensional decomposition.
> > >
> > > We agree that some conditional distributions are not defined on normed vector spaces. We will make the claim more precise and include a more detailed discussion on this point in our final version.
> > >
> > > [1] Tongzheng Ren, et al. Stochastic nonlinear control via finite-dimensional spectral dynamic embedding. 2023.
> > >
> > > **W2.** We have clarified the SVD in the above question. For further references about the property of linear MDP, please refer to [1].
> > >
> > > [1] Chi Jin, et al. Provably Efficient Reinforcement Learning with Linear
> > > Function Approximation. 2019.
> > >
> > > **W3.** Among the references attached with the last question, [1] and [2] demonstrate that when the ground-truth feature vector $\phi^*$ or the kernel function $k$ is given, efficient learning can be achieved by incorporating a UCB-style bonus based on $\phi^*$ or $k$. In practical cases where such features are not known a priori, methods like [3] and [4] incorporate representation learning to obtain such features, although with a deteriorated regret bound compared to [1]. We acknowledge that certain claims, such as "learning xxx requires xxx" may be too definitive, and we will adopt a more nuanced expression as you suggested.

---

### Official Review · Reviewer_2EpY · 2024-07-08

**Soundness:** 3
**Presentation:** 3
**Contribution:** 3
**Rating:** 6
**Confidence:** 3

**Summary:**

This paper proposes Diffusion Representation (Diff-Rep). This approach leverages diffusion models to learn representations for value functions in reinforcement learning while avoiding the high inference cost of sampling from diffusion models.

**Strengths:**

1. Diff-Rep provides a novel and principled approach to leveraging the flexibility of diffusion models for reinforcement learning without incurring their high inference costs. The key technique used here is exploiting the energy-based model view of diffusion models.

2. Empirically, this approach demonstrates solid performances across various continuous control tasks. Diff-Rep consistently outperforms competitive baselines, including prior diffusion-based RL methods and state-of-the-art model-based and model-free algorithms.

3. The writing of this paper is smooth and easy to follow.

**Weaknesses:**

1. While the paper provides some results on the expressivity of the learned diffusion representations, the theoretical analysis is still somewhat limited. More formal characterizations or guarantees on the representation power, such as error bounds on the Q-function approximation, would further strengthen the technical contributions.

2. The experimental evaluation, while extensive, focuses primarily on simulated benchmarks. The lack of more realistic or practical tasks is a disadvantage. Such results could help illustrate the approach's scalability and robustness under more complex and noisy conditions, which cannot be evaluated in a relatively easy environment.

**Questions:**

1. As mentioned, regarding why the diffusion representations can expressively approximate the Q-function, are there any theoretical results or analyses that could formalize this representation power more precisely? What are the potential difficulties in providing such results?

2. What are some example real-world applications where you think Diff-Rep could provide the most benefit over prior approaches? Do you consider such extensions for the next/future version?

3. Optimizing the representation objective (18) alongside the model-free RL updates seems burdensome. Could you comment on the computational/sample complexities of training this? Moreover, did you explore any other training strategies, such as pre-training the representation? I guess it necessarily means some trade-offs. Adding discussing and comparing the tradeoffs here might be interesting.

4. While this paper mainly studies Diffusion for RL, the energy-based formulation and the KL-based training objectives presented in this paper are somewhat similar to those of papers studying (KL-regularized) RL for Diffusion (e.g., [1-2]). It might be interesting to discuss the methodological relations/differences between the two interdisciplinary directions.



[1] https://arxiv.org/abs/2305.16381

[2] https://arxiv.org/abs/2402.15194

**Limitations:**

The authors adequately addressed limitations.

---

> ### Author Rebuttal · Authors · 2024-08-07
>
> * **W1 \& Q1: Limited theoretical analysis.**
>
> The major contribution of this paper is developing an efficient algorithm for sufficient representation through diffusion, with intensive empirical evaluation.
> We would like to emphasize that our paper aligns with the existing algorithms [1][2] in leveraging the spectral representations for RL. The difference lies in that our approach efficiently extracts and constructs such representations via diffusion. Therefore, our method enjoys the same theoretical guarantee concerning sample complexity (e.g., Theorem 9 in [2]). In the next revision, we will include a remark that elaborates on the relationship between Diff-Rep and prior works, and refer interested readers to the theoretical analysis.
>
> [1] Tongzheng Ren, et al. Latent variable representation for reinforcement learning. 2022.
>
> [2] Tongzheng Ren, et al. Spectral decomposition representation for reinforcement learning. 2022.
>
> * **W2 \& Q2: Lack of realistic or real-world tasks.**
>
> The primary objective of this research is to introduce a diffusion-based method for learning spectral representations. We followed the standard comparison protocol in [1,2] for both MDP and POMDP settings for fair comparison. The improvements in simulations demonstrate the effectiveness of our approach as our first step.
>
> We are developing our method for applications in more realistic tasks as our future research. Given the capability of our method to extract sufficient representations for subsequent tasks (e.g. representing value functions), we envision its broader applications such as multi-task robotics control and preference alignment for large language models.
>
> [1] Tongzheng Ren, et al. Latent variable representation for reinforcement learning. 2022.
>
> [2] Hongming Zhang, et al. Provable Representation with Efficient Planning for Partially Observable Reinforcement Learning. 2023.
>
>
> * **Q3: Optimizing objective (18) along with model-free RL seems burdensome and other training strategies, such as pre-training the representations, may be exploited.**
>
> In our implementation, we built $Q$-functions on top of the representation network $\psi$, and used objective (18) to train the representation layers. Therefore, Diff-Rep incurs minimal additional costs compared to model-free algorithms. On the other hand, conventional model-based methods such as MWM involve independent representation learning, dynamics modeling and actor-critic learning processes, while our method leverages the benefits of dynamics from a representation learning perspective, thereby significantly reducing training costs.
>
> In this paper, we mainly focus on the online RL settings to follow the evaluation protocols of existing representation-based RL methods. As the data is progressively collected by the agent, pre-training the representations beforehand is not feasible due to insufficient data. However in offline scenarios where extensive datasets are accessible, pre-training the representations is likely to bring more stable and improved outcomes.
>
>
> * **Q4: Connections to papers about RL for Diffusion.**
>
> Some studies within these two fields are connected in terms of methodology. Suppose the diffusion model is pre-trained on the data with distribution $p_{\text{data}}(x)$, both fields generally aim to sample from an enhanced distribution w.r.t. some evaluation metric $Q(x)$:
> $$p_{\text{target}}(x)\propto p_{\text{data}}(x)\exp(Q(x)).$$
> In Diffusion for RL, typically $x$ is the trajectory $\tau$ while $Q(x)$ is the overall return of the trajectory $R(\tau)$. In the application of RL to Diffusion, $Q(x)$ can manifest as some specific metric, such as the aesthetic quality in image generation or bioactivity in biological sequence generation, as ELEGANT did.
>
> Despite this, our method leverages the capabilities of diffusion models from the perspective of representation learning, which fundamentally differs from the aforementioned methods.

---

> > ### Comment · Reviewer_2EpY · 2024-08-09
> >
> > Thanks for addressing my questions. The impact of this work can be largely promoted if applied on multi-task robotics control and preference alignment for large language models. But I acknowledge the empirical contributions made by the authors. I tend to keep the current score.

---

### Official Review · Reviewer_TNcM · 2024-07-10

**Soundness:** 3
**Presentation:** 3
**Contribution:** 3
**Rating:** 7
**Confidence:** 2

**Summary:**

The paper proposed a representation learning method based on diffusion model. The paper developed the method using the EBM setting of the transition probability, and proposed a finite dimension approximation of the state-action representation by minimizing an orthormal regularization term. The performance of the approach is supported with various experiments.

**Strengths:**

The paper has sufficient comparison with the existing works. The writing is clear and easy to follow. The derivation of the diffusion setting from EBM is novel and the experiment results are convincing.

**Weaknesses:**

To make the paper more complete, the authors can make some discussions on the representation quality, such as running the algorithm on a toy latent-state-MDP and show if the algorithm is able to find the latent state-representation.

**Questions:**

As the representation learning enables the linear representation of the Q-value, one would have closed-form solution to Equation (20). On the other hand, the algorithm still uses gradient descent with the double-Q-network trick. Is there any specific reason of doing so?

**Limitations:**

/

---

> ### Author Rebuttal · Authors · 2024-08-07
>
> * **W1: Some discussions on the representation quality are expected, e.g. to check whether Diff-Rep can recover latent state representations.**
>
> We would like to emphasize that Diff-Rep focuses on extracting representations that are sufficient to represent the $Q$-functions, rather than the latent representations of $(s, a)$. Therefore, the representations from Diff-Rep are generally different from the latent state of the MDP. To illustrate the representation quality, we included the generation results and also conducted experiments in a toy MDP where the ground truth $Q$-function can be solved for analysis. Please refer to Appendix F.2 and the general response for details.
>
> * **Q1: Why use gradient descent rather than closed-form solution?**
>
> While it is possible to optimize the $Q$-value functions through least squares regression, this approach would incur significant computational costs such as inverting the covariance matrix. Instead, we opt for gradient descent to optimize the $Q$-value functions for its simplicity and efficiency in our online setting. Note that when the representations are fixed, the objective is convex, and thus gradient descent gives the same solution as the closed-form solution.

---

### Author Rebuttal · Authors · 2024-08-07

We would like to thank all our reviewers for their effort and time in providing constructive suggestions for our paper. We are delighted that our reviewers recognized the relevance of our problem and the novelty of our method.

We note that both the reviewer adAM and iUqH raised concerns regarding the expressiveness of Diff-Rep in representing the $Q$-functions of any policy $\pi$. Therefore, we conduct experiments with a toy environment, FrozenLake-v1 (\url{https://www.gymlibrary.dev/environments/toy_text/frozen_lake/}). FrozenLake is a grid world where an agent navigates a frozen lake from the Start (S) to the Goal (G) while avoiding Holes (H). The environment features a 4x4 discrete state space and a discrete action space with 4 possible movements: left, down, right, and up. The agent's movement can be affected by the slippery surface, adding stochasticity to the transition. Rewards are given for reaching the goal (+5) or the holes (-1).

We first train an optimal policy in this environment, and then at each step, we let the policy take random actions with different probabilities $\epsilon_i\in\\{0,0.2,0.4,0.6,0.8,1.0\\}$ to simulate different policies $\pi_i$. Then we conduct policy evaluation to obtain ground truth values of $Q^{\pi_i}$.

Our objective is to find out whether the diffusion representation in our paper can sufficiently represent each $Q^{\pi_i}$ with sufficiently low error, as our theory suggests. The diffusion representation proposed in our paper is given by $\phi(s, a) = \texttt{elu}(W\sin(\Omega^\top \psi(s, a)))$, where $\mathbb{P}(\tilde{s}'|s, a; \beta)\propto \exp(\psi(s, a)^\top \nu(\tilde{s}, \beta))$, $\Omega$ is composed by $\omega\sim\mathcal{N}(0, I)$, and $W$ is a learnable matrix. To obtain such representations, we treat the coordinates of the grid as continuous values and perform diffusion training based on the coordinates. Once we obtain $\psi$, we can construct the diffusion representations $\phi$, fix them, and use them to regress the ground truth $Q^{\pi_i}$.

The residual errors of the regression are listed in the following table. As an approximation to the spectral representations, Diff-Rep can accurately represent $Q$-values of any policy $\pi_i$, which validates its expressiveness of representing $Q$-values.

|$Q$-function|$Q^{\pi_0}$|$Q^{\pi_1}$|$Q^{\pi_2}$|$Q^{\pi_3}$|$Q^{\pi_4}$|$Q^{\pi_5}$|
|---|---|---|---|---|---|---|
|Residual Error|$1.2\times 10^{-8}$|$3.6\times 10^{-7}$|$1.2\times 10^{-7}$|$4.9\times 10^{-9}$|$2.0\times 10^{-8}$|$9.2\times 10^{-8}$|

---

### Decision · Program_Chairs · 2024-09-25

**Decision:**

Accept (poster)

**Comment:**

The paper presents an interesting new idea by observing that:

1. it is well known that better representations can lead to better RL
   performance.
2. diffusion models are well recognized for their ability to represent
   high-dimensional multi-modal distributions and that this can translate
   into SOTA RL performance in some cases.
3. however, standard use of diffusion models in RL requires computationally
   expensive sampling which limits their practical use.

It then proposes a new representation learning method that uses insights
and loss functions adapted from diffusion models along with random fourier
features.  The representation learning is integrated into a relatively
standard RL algorithm that learns value functions and policies as linear
functions of the learned representation.  The method is compared against a
variety of diffusion-based and more traditional deep RL methods on 10
standard RL benchmarks with the proposed method performing at or above all
other methods on most of them.  There are POMDP examples in the appendix
which are presented more like examples rather than comparisons.

Note that the MDP comparison's training iterations are capped before
training has converged in any of the methods.  This shows the sample
efficiency of the early learning phase but gives no insight into the
converged performance of any of the methods.  Although this practice is not
uncommon in the RL literature, it does limit the evaluation.  Ideally,
comparisons at convergence would also be included.

Overall, the paper proposes an interesting new algorithm with good insights
from recent literature and strong empirical results against a broad set of
comparison algorithms.